

# Generating wind power scenarios for probabilistic ramp event prediction using multivariate statistical post-processing

Rochelle P. Worsnop[1], Michael Scheuerer[2,3], Thomas M. Hamill[3], Julie K. Lundquist[1,4]

[1]Department of Atmospheric and Oceanic Sciences, University of Colorado Boulder, Boulder, Colorado, USA
[2]Cooperative Institute for Research in the Environmental Sciences, University of Colorado Boulder, Boulder, Colorado, USA
[3]NOAA/ESRL, Physical Sciences Division, Boulder, Colorado, USA
[4]National Renewable Energy Laboratory, Golden, Colorado, USA

*Correspondence to*: Rochelle P. Worsnop (Rochelle.worsnop@colorado.edu)

**Abstract**

Wind power forecasting is gaining international significance as more regions promote policies to increase the use of renewable energy. Wind ramps, large variations in wind power production during a period of minutes to hours, challenge

utilities and electrical balancing authorities. A sudden decrease in wind energy production must be balanced by other power generators to meet energy demands, while a sharp increase in unexpected production results in excess power that may not be used in the power grid, leading to a loss of potential profits. In this study, we compare different methods to generate probabilistic ramp forecasts from the High Resolution Rapid Refresh (HRRR) numerical weather prediction model with up to twelve hours of lead time at two tall-tower locations in the United States. We validate model performance using 21

months of 80-m wind speed observations from towers in Boulder, Colorado and near the Columbia River Gorge in eastern Oregon.

We employ four statistical post-processing methods, three of which are not currently used in the literature for wind forecasting. These procedures correct biases in the model and generate short-term wind speed scenarios which are then converted to power scenarios. This probabilistic enhancement of HRRR point forecasts provides valuable uncertainty

information of ramp events and improves the skill of predicting ramp events over the raw forecasts. We compute Brier skill scores for each method at predicting up- and down- ramps to determine which method provides the best prediction. We find that the Standard Schaake Shuffle method yields the highest skill at predicting ramp events for these datasets, especially for up-ramp events at the Oregon site. Increased skill for ramp prediction is limited at the Boulder, CO site using any of the multivariate methods, because of the poor initial forecasts in this area of complex terrain. These statistical methods can be

implemented by wind farm operators to generate a range of possible wind speed and power scenarios to aid and optimize decisions before ramp events occur.





## 1 Introduction

Global wind-energy installation reached 486 GW in 2016; the total installed generation capacity in the US alone
reached > 82 GW by the end of 2016 and has experienced a rapid rise since then (GWEC, 2017). Increased interest in alternatives to fossil-fuel-based energy to mitigate greenhouse gas emissions as outlined in the international Paris Agreement (UNFCCC, 2015) has propelled the global wind-energy sector even further. Because of increased interest and deployment of wind energy in the US and worldwide, accurate wind speed and power forecasts are becoming increasingly important for successful power-grid operation. In particular, the prediction of specific wind situations such as power ramps is key to
effective operation and control of wind farms (Kuik et al., 2016).

Power ramp events are challenging to forecast because these abrupt and large increases or decreases in wind speed - and thus power - happen on time scales of minutes to hours making it difficult for wind farm operators and the power grid to respond. Up-ramps, or sharp increases in wind-farm power can lead to an overload of electricity generation. Sometimes the additional electricity is sold to nearby utility companies, but frequently wind farms must curtail, or stop power production if
there is not enough time to make the sale. Conversely, down-ramps, or sharp decreases in power production over short time periods, also have serious implications for the power grid. If power generation from the wind farm does not meet contractual expectations, then power must be generated by another source to "balance the load" and avoid brownouts and blackouts. Additionally, the wind farm owners may have to pay costly fees for not meeting their quota.

Improving the accuracy of ramp forecasts can help avoid the situations described above. The overall effects of
ramps on the grid can be reduced in several ways. The development of a geographically aggregated power grid which connects many wind farms and diverse renewable sources such as solar, hydro, and nuclear power (Budischak et al., 2013) can help minimize the effects of sudden gusts and lulls of wind speed on the power grid. Additionally, optimized wind farm locations and layouts (St. Martin et al., 2015) could reduce fluctuation on the grid caused by individual wind farms. Directly improving ramp forecasts is also a viable option to reduce stress on the power grid and make wind energy even more
reliable. Increased reliability may be realized in the form of decision-making. A wind farm operator may make conservative estimates of how much power their wind turbines can generate during times with an elevated probability of a down-ramp event. In practice, a persistence forecast of wind speed and power generation over a 1-h or 30-min time interval is commonly used (Milligan et al., 2003). Persistence forecasts are generally reasonable on these time scales, because local weather conditions usually do not change drastically during these lengths of times except during certain weather events, such as
fronts, convective outflow, etc. that often cause ramps. However, persistence forecasts are poor at predicting ramps; a ramp identified in the previous 30 min to an hour can change magnitude or even sign (i.e., up- or down-ramp) in a short period and therefore lead to large forecast errors. In recent years, there has been a growing interest in information regarding the uncertainty of wind power forecasts to make energy decisions (Nielsen et al., 2006b). Typical single (i.e., point) forecasts cannot provide this necessary uncertainty information, but probabilistic forecasts can.




Considerable effort over the last decade has been made to improve short-term wind and power forecasts (Wilczak et al., 2014). To improve beyond the use of persistence of a point forecast, some of these methods include the use of predictive distributions broken into quantiles for each lead time to quantify uncertainty. These methods neglect the serial correlation among forecast lead times (Bremnes, 2006), a characteristic needed for time-dependent events such as the evolution of ramps. Other methods construct the serial dependence across forecast lead times, but achieve the original quantiles (i.e.,

margins) from non-parametric forecast distributions, which make no assumptions about the shape of the actual forecast distribution (Pinson et al., 2009; Pinson and Girard, 2012). Another method includes the direct use of an ensemble of forecasts produced by perturbing the initial conditions of a numerical weather prediction (NWP) model, which does not require the generation of predictive distributions and their serial correlation across lead times through statistical means. However, the ensembles themselves are under-dispersive and lack small-scale variability in time and space so that not all

possible scenarios are captured (Nielsen et al., 2006a; Bossavy et al., 2013). Others have used analogs of past forecasts based on weighted atmospheric predictors to quantify forecast uncertainty (Delle Monache et al., 2013; Junk et al., 2015).

        In the research to be discussed here, we will correct biases in wind speed point forecasts produced by the HRRR NWP model using univariate post-processing techniques and parametric distributions. We will then test four multivariate statistical post-processing methods to generate forecast scenarios of wind speed, representing the prediction uncertainty for a

12-h forecast horizon. We then compare the skills of the methods at predicting up- and down-ramp events. Three of the four methods, (the standard Schaake Shuffle (StSS), Minimum Divergence Schaake Shuffle (MDSS), and the enhanced version of MDSS (MDSS+)) are not currently discussed within the wind-forecasting literature and are offered as new forecasting tools for short-term ramp events. The fourth method, the Gaussian copula, has been assessed for short-term wind and power forecasting, so we use this method as a benchmark of performance for the new methods. For all of our analyses, we

physically compute wind power production via a turbine power curve, which relates the power that would be generated by a turbine to wind speed through the turbine rotor layer as well as turbine-specific characteristics.

        The wind speed observations from tall meteorological towers and forecasts from the HRRR model used in this study are discussed in Sect. 2. Up- and down-ramp events are formally defined in Sect. 3.1. Univariate post-processing of the raw HRRR forecasts is described in Sect. 3.2. The multivariate methods for generating probabilistic forecast scenarios are

discussed in Sect. 3.3. In Sect. 4, we evaluate the performance of each probabilistic forecast method and the raw HRRR forecasts focusing on the prediction of up- and down-ramp events. Specifically, we compare the relative frequency of up- and down-ramp events produced from each forecast. We also provide Brier skill scores to compare each multivariate method and to show the performance relative to climatology. In Sect. 5, we offer concluding remarks, uses for the probabilistic methods in the wind-energy sector, and advice for operational implementation.




## 2 Data

### 2.1 Wind measurements from tall meteorological towers

We use wind speed and direction measurements from two meteorological towers. The first tower is the 135-m M5 tower located south of Boulder, Colorado and ≈ 5 km east of the Colorado Front Range at the US Department of Energy's (DOE)
National Wind Technology Centre (NWTC) (Clifton et al., 2013). Wind speed and direction measurements from the M5 tower were collected at 80 m and 87 m above ground level (a.g.l), respectively, from a cup anemometer and wind vane. The instruments were mounted on tower booms aligned at 278°, the prevailing wind direction at the NWTC based on a 15-yr climatology (Clifton et al., 2013). We remove wind speed measurements that are associated with wind directions between 75° −135° to ensure that the measurements are not contaminated from the flow passing through and around the tower or
waked by a nearby wind turbine before reaching the instrument sensors. We also remove data flagged by quality-control methods such as testing for constant values during a measurement interval (which indicates icing events during cold months), and checking for standard deviation values < 0.01% of the mean (which indicates instrument malfunction) among other measures described by Clifton et al. (2013) and St. Martin et al. (2016). After filtering, 81% of the data were retained. The M5 tower data that we use are measured at a 20-Hz rate and averaged over ten minutes for the period from 31 August
2012 to 28 February 2017.

The second tower is an 80-m tall proprietary tower located near the Columbia River Gorge, which divides the southern boundary of Washington and the northern boundary of Oregon. Herein, we refer to this tower as the Pacific Northwest (PNW) tower. The wind speed and direction measurements are collected from a heated cup anemometer and wind vane at 79 m and 76 m a.g.l, respectively and averaged to 1 min. We perform quality-control measures on the data to remove
suspect data using similar quality-control processes as for the M5 tower. We also remove unrealistic wind speed values, such as negative numbers, and remove data associated with waked flow from the PNW tower or nearby turbines. After filtering, 73% of the data were retained. Data from the PNW tower were made available as part of the DOE-funded second Wind Forecast Improvement Project (WFIPII) that took place from fall 2015 to spring 2017 (A2E, 2017). We use data from this tower for all available dates between 18 March 2015 − 06 March 2017.


### 2.2 Wind forecasts from HRRR system

Deterministic forecast data are obtained from the second experimental version of NOAA's real-time, High-Resolution Rapid Refresh assimilation and model forecast system (HRRRv2). The HRRRv2 domain covers the contiguous US at 3-km horizontal resolution. HRRRv2 is updated hourly with initial conditions from the 13-km Rapid Refresh model and
observations via data assimilation. Detailed model physics for HRRRv2 is discussed by Benjamin et al. (2015). The available dates for this version of the HRRR are from 01 January 2015 − 28 September 2016. Forecast verification is



performed on this period of interest which overlaps with the observation availability. For comparison of the 80-m wind speed forecasts to the tower observations, the HRRR forecast values at each tower location are from the nearest model grid cell to the tower latitude and longitude. In addition, since the HRRRv2 forecasts are output hourly, we apply our analyses to

the observations that occur at the top of the hour to match the forecast availability. For the observations and model output, we only analyse dates that have a continuous 12-h segment of data from the 00Z and 12Z forecast initialization times to encompass an entire day. These criteria yield ≈ 80–150 continuous 12-h forecast segments that overlap with available observations for each initialization time and tower location. The criteria also yield ≈ 300–400 continuous 12-h segments of observations for each initialization time and tower location that will be used for forecast verification and the multivariate

methods discussed in Sect. 3.3.

## 3 Methods

### 3.1 Ramp definition

Wind power ramps are large changes in power production over short time periods. Despite the significant influence of ramp events on the electric grid and a clear need for accurate forecasts of these events, there is not a commonly accepted method

to define and identify them. Ramp definitions vary in the literature (Kamath, 2010, 2011; Pinson and Girard, 2012; Bossavy et al., 2013; Bianco et al., 2016) regarding the threshold of power change and the duration over which that change occurs. Variations also exist regarding which data points in a given window of time should be used when calculating the change in power, and lastly whether to use power time series directly when defining ramps or instead use a filtered time series (Bossavy et al. 2013). Commonalities in the literature include the need to define ramp magnitude, duration, and sign (i.e., up-

or down-ramp).

This lack of a standard definition is primarily because what is considered an important ramp event depends on the needs of the wind farm operator or grid-system manager at any given time or location. Here, we employ a combination of the *Minimum-maximum* method used by Pinson and Girard (their eqn. 8, 2012) and that employed in the Ramp Tool & Metric created by Bianco et al. (2016) to generate separate ramp time series for up-ramps and down-ramps. Up-ramps and down-

ramps are considered separately, because they have different impacts on the power grid and lead to different decisions. Up-ramps may result in a swap of conventional energy sources for cleaner wind power while a down-ramp may result in the opposite and can have more detrimental effects on the grid during periods of high electricity demand.

Before identifying power ramps, wind speed observations and forecasts must be converted to power. A conversion from wind speed to power in this study is achieved via the International Electrotechnical Commission (IEC) turbine power

curve for Class 2 turbines (IEC, 2007). This power curve is for wind turbines with a cut-in wind speed > 3 m s$^{-1}$, rated power ≥ 16 m s$^{-1}$, and a cut-out wind speed > 25 m s$^{-1}$. Using the resulting power time series, we create binary time series of up-

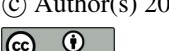



and down-ramp events into ones (ramp occurred) and zeros (no ramp occurred). We do this by first dividing the power time series into $N_{win}$ sliding time windows of length $h$ and then finding the largest positive and negative power differences that exists within each window, $\Delta p_{max}$ and $\Delta p_{min}$, respectively. If the largest positive power difference equals or exceeds the

defined power change threshold $\xi$, then the up-ramp time series is given a value of 1 for that time window. Conversely, if the largest negative power difference is less than or equal to $-\xi$, then a 1 is assigned to the down-ramp time series for that time window. If the above respective criterion is not met, then a 0 is assigned for that time window. The window then slides one hour forward in time and the process is repeated until there are $N_{win}$ binary values for both the up-ramp and down-ramp time series. We allow up-ramps and down-ramps to happen within the same time window, so that there could be a value of 1

assigned for the same time window in both the up- and down-ramp time series. This allowance is reasonable, because for some longer window-lengths, up-ramps and down-ramps could both occur and are equally important to forecast. An example of the identification of up- and down-ramps according to this method appears in Figure 1. While more complex ramp definitions are available, the chosen criteria for up- and down-ramps reflect the common intuition about ramps including threshold $\xi$ and window length $h$ to customize the definition to specific needs. As determined later, this ramp definition can

be employed in a probabilistic framework and will be used to compare the different approaches to scenario generation.





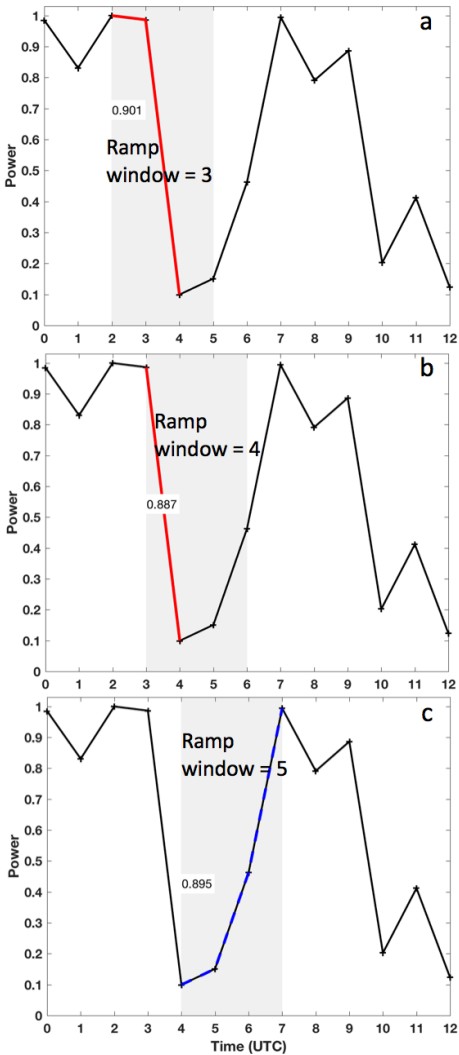

**Figure 1. Ramp identification at the M5 tower location for an observed power time series at 00Z on 03 March 2016 for a window**
**size $h$ = 3 h and change in power threshold $\xi$ = 60% of turbine power capacity. Three consecutive time windows are shown as the**
**grey rectangles in a, b, and c. The identified ramps in those time windows are highlighted in red (up-ramps with ≥ 60% power**
**change) and blue (down-ramps with ≤ −60% power change). The change in power capacity associated with each ramp is written**
**in the white textboxes within the grey time windows. The total number of up- and down-ramps identified within all 3-h sliding**
**time windows is 2 and 6, respectively.**



### 3.2 Deterministic to probabilistic forecasts: univariate post-processing

To improve the skill of the raw HRRR forecasts at predicting ramp events, we employ statistical post-processing techniques to enhance the HRRR forecasts through the addition of uncertainty information. These methods convert the deterministic (single value) raw HRRR forecast into probabilistic forecasts by creating a set of forecast scenarios of wind speed that represent the forecast uncertainty. Wind speed scenarios are converted to power scenarios and then probabilities of ramp events are derived. The first step to generating scenarios is to perform univariate post-processing on the HRRR forecasts at

each individual lead time.

We first determine a predictive distribution model for each tower and forecast initialization time which accurately predicts future observations for each forecast lead time. We employ ordinary-least-squares regression on the observed wind speed data during which the HRRR forecasts are also available (01 January 2015 – 28 September 2016). To make use of the ≈ 21 months during which both the HRRR forecasts and observations are available, we cross-validate these data. We leave

one month out for verification and fit the statistical models used to determine the parameters of the predictive distributions with the remaining 20 months of data (training period). We repeat this process so that 21 months of forecasts and independent verifying observations are obtained for each month, forecast initialization, lead time, and tower location. We find the mean and standard deviation of the predictive distributions by inserting verifying forecasts into the fitted regression model. Before performing the regression, we apply a power transform (not to be confused with wind speed-to-power

conversion) with power exponent $P$ to the forecasts $\tilde{x} = x^P$ and observations $\tilde{y} = y^P$ to address the increase of forecast uncertainty with wind speed (i.e., heteroscedasticity in the dataset). Heteroscedasticity in the data is visible as more spread in the data points at higher wind speeds than at lower wind speeds in Figure 2a. We select power exponents for the transformations that produce slope coefficients nearest to zero from a second regression of the absolute residuals from the first regression on the transformed forecasts. The exponent is 0.66 (0.75) for both forecasts and observations at each

initialization time and all lead times at the NREL M5 (PNW) tower.

After applying a power transform to the data, we remove the seasonal cycle for each location, initialization time, and lead time by normalizing the transformed forecasts and observations by the corresponding seasonal cycle. The seasonal cycle model takes on the form,

$$s(T) = a_0 + a_1 \sin(2\pi T) + a_2 \cos(2\pi T) \qquad (1)$$

and the model coefficients $a_0$, $a_1$, and $a_2$ are determined by fitting the seasonal cycle model to the transformed forecasts for every forecast date in the form of fractional day of the year $T$. We fit the seasonal cycle model solely on the transformed forecasts, because there are more forecasts than observations available during the period of interest. Therefore, the same

seasonal cycle coefficients are used to derive the seasonal cycle for the transformed forecasts and observations.



The transformation and removal of the seasonal cycle makes the relationship between the transformed forecasts and transformed observations more homoscedastic (i.e., more consistent forecast variability for all wind speeds in Figure 2b). This characteristic enables us to use more traditional statistical techniques and avoid the nonlinear regression that would be required between the untransformed forecasts and untransformed observations because of the heteroscedasticity. An inverse transformation of the observations, forecasts, and regression lines reveal the complexity of the regression line we would have had to use if we had not transformed the data before applying regression analysis (Figure 2c). The slight curvature in the standard deviation lines in Figure 2c shows the dependence of error variance on wind speed magnitude (i.e., heteroscedasticity); the black dots are closer to the red regression line at lower wind speeds than at higher wind speeds. The scatter in the red regression dots in Figure 2c illustrates how the annual cycle influences the regression; depending on the time of year, the transformation value can be different because of the annual cycle.

We test three candidate predictive distribution models for the transformed wind speed observations using the predictive mean and standard deviation produced from the above linear regression: truncated normal, truncated logistic, and gamma distributions where the truncated distributions exclude negative values. These distributions, given the same mean and standard deviation, vary in the shape of their peaks and size of their tails. Probability integral transforms (PITs) of each predictive cumulative distribution function (CDF, $F_i$) and its verifying observation $y_i$ are calculated for each candidate distribution as $d_i := F_i(y_i)$, and provide an assessment of which distribution yields the best calibration (Dawid, 1984; Gneiting et al., 2007). Histograms of the PITs which include all verification days and lead times show that the gamma and truncated logistic distributions are well-calibrated to the observed transformed wind speeds at the NREL M5 and PNW towers, respectively for 00Z (Figure 3) and 12Z (not shown) initialization times. The good calibration is qualitatively demonstrated by the mostly uniform histograms in Figure 3. For a more quantitative assessment of calibration, we compute the continuous ranked probability score (CRPS). The CRPS is a proper scoring rule that is often used to evaluate the quality of a probabilistic forecast by summarizing the sharpness and calibration of the forecast distribution (Gneiting et al., 2005; Gneiting and Raftery, 2007). A proper score is one that produces the highest reward (i.e., lowest CRPS score) by using the true probability distribution (Gneiting and Raftery, 2007). For a given pair of predictive CDF $F$ and verifying observation $y$, the CRPS is defined as

$$\text{CRPS}(F, y) = \int_{-\infty}^{\infty} [F(\xi) - \boldsymbol{H}(\xi - y)]^2 d\xi, \tag{2}$$

where $F(\xi)$ is the probability that the forecast will not exceed threshold $\xi$ and $\boldsymbol{H}$ is a Heaviside step function which attains the value 1 when its argument is $\geq 0$, and 0 otherwise. A low CRPS value suggests a predictive distribution model can accurately represent future observations. We calculate the CRPS for each candidate predictive distribution using the closed-form expressions for the CRPS of a truncated normal (Gneiting et al., 2006) and the truncated logistic and gamma


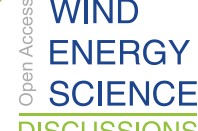


distributions (Scheuerer and Möller, 2015). Based on the CRPSs, averaged over all lead times for each tower and initialization (Table 1), and the PIT histograms, we choose to proceed with the gamma (truncated logistic) distribution model

for the NREL M5 (PNW) transformed observations.

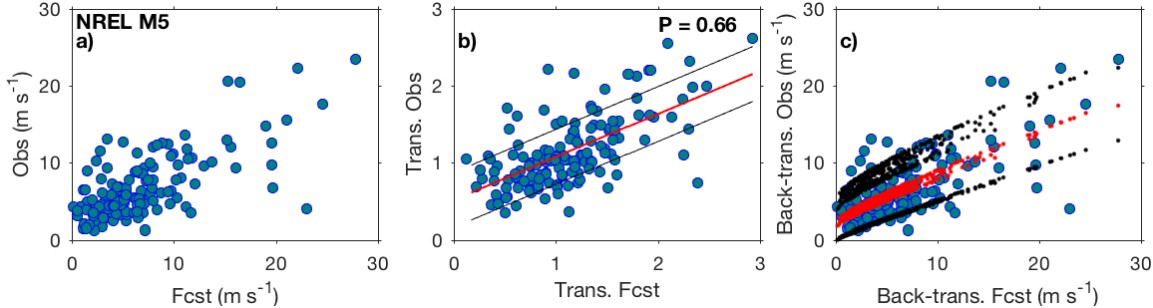

**Figure 2.** Scatter-plots of (a) observations (Obs) versus forecasts (Fcst), (b) unit-less transformed observations versus transformed forecasts, and (c) the back-transformation of the observations versus the back-transformed forecasts from the NREL M5 tower at
an 00Z initialization and a two-hour forecast lead time. The exponent P used in the power transformation is shown in (b). The least-squares linear regression trends (solid red line in (b) and red dots in (c)) and lines representing one standard deviation (solid black line in (b) and black dots in (c)) from the regression lines are displayed for the transformed and back-transformed data.



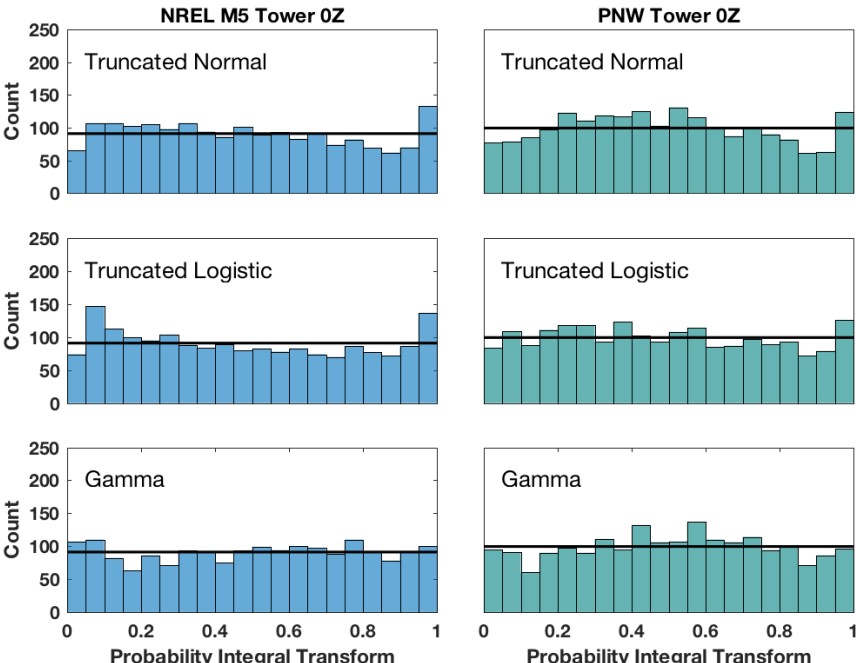


**Figure 3 Histograms of the probability integral transform (PIT) using the predictive truncated normal, truncated logistic, or gamma distribution models at 00Z for the M5 tower (blue) and PNW tower (green). The horizontal black line depicts the count that each of the twenty bins would have if the histogram was perfectly uniform.**

**Table 1. Average CRPS (in m s⁻¹) for the 00Z and 12Z calibrated probabilistic forecasts obtained using the truncated normal ($\mathcal{N}_0$),**
**truncated logistic ($\mathcal{L}_0$), and gamma ($\mathcal{G}$) predictive distribution models. The arrow represents the direction for good scores and the best scores are shown in bold.**

|  |  | NREL M5 | PNW ↓ |
|---|---|---|---|
| $\mathcal{N}_0$ | 00Z | 0.203 | 0.137 |
|  | 12Z | 0.234 | 0.157 |
| $\mathcal{L}_0$ | 00Z | 0.203 | **0.136** |
|  | 12Z | 0.235 | **0.156** |
| $\mathcal{G}$ | 00Z | **0.202** | 0.137 |
|  | 12Z | **0.233** | 0.158 |

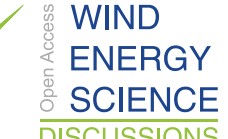

### 3.3 Generation of forecast scenarios: multivariate post-processing

We obtain probabilistic forecasts of univariate statistically post-processed wind speeds for each verification day, forecast
initialization, and lead time for both towers by using the truncated logistic or gamma distribution models as discussed in
Sect. 3.2. These marginal distributions provide prediction uncertainty information for each lead time on a given day and
initialization time, but they do not provide information about the serial dependence of the distributions across multiple lead
times. Ramp events are changes in power over a short period of time; to identify ramps and the uncertainty associated with
them, we need to generate scenarios of wind speed which represent that serial dependence and that can then be converted to
scenarios of wind power. We model serial dependence of the individual lead-time predictive distributions to construct
forecast scenarios of wind speed which are then converted to power. We utilize four methods to define the interdependence
structure and generate the scenarios. The Gaussian copula, standard Schaake Shuffle (StSS), MDSS, and MDSS+ methods
are discussed below.

### 3.3.1 Gaussian copula

We first generate scenarios of wind speed following the Gaussian copula method (Pinson et al., 2009; Pinson and Girard,
2012). The Gaussian copula approach first converts the transformed wind speeds (Sect. 3.2) from the chosen forecast
distribution (here, we use truncated logistic or gamma) into a uniform marginal probability distribution and then converts the
uniform values into standard Gaussian-space using a combination of CDFs $F_{\mathcal{D}}$ and inverse CDFs $F_{\mathcal{D}}^{-1}$, where $\mathcal{D}$ is either a
gamma $\mathcal{G}(\lambda, r)$, truncated logistic $\mathcal{L}_0(\mu, \sigma)$, or Gaussian $\mathcal{N}(0,1)$ distribution. A flow diagram of the Gaussian copula
procedure starting with a marginal gamma distribution is shown in Figure 4 and described below. An empirical covariance
matrix of the Gaussian values is constructed to estimate the correlation between the Gaussian values from all pairs of lead
time. This covariance matrix provides information necessary to transition from marginal distributions for each lead time to
multivariate distributions, which inform how the Gaussian values link across multiple lead times. Given the limited amount
of training data and gaps in the range of dates for which observations are available, we do not attempt to estimate a time-
varying covariance model. Instead, we follow Pinson and Girard (2012) and use a fixed exponential covariance model
(ECM),

$$ECM(X_{k1}, X_{k2}) = \exp\left(-\frac{|k_1 - k_2|}{\nu}\right), \tag{3}$$

where $X_{k1}$ and $X_{k2}$ are the Gaussian random variables at lead time $k_1$ and $k_2$, respectively, and $\nu$ is the range parameter
which controls the extent of correlation of transformed wind speed across lead times. An appropriate value for $\nu$ is selected
empirically so that the resultant ECM for a given value of $\nu$ most resembles the decay of the empirical correlation values
(Appendix A). A random number generation of the ECM for a given $\nu$ is employed to generate scenarios of multivariate



Gaussian-distributed values. Those Gaussian-distributed scenarios are then converted to scenarios with uniform margins by

taking the CDF of a standard Gaussian distribution evaluated at the Gaussian-distributed values. An inverse CDF of the forecast marginal distribution (here, we use truncated logistic or gamma) of the uniform values yields the final result of transformed wind speed scenarios with marginal distributions as determined in Sect 3.2. For this study, we generate 1000 Gaussian copula scenarios of transformed wind speed. We then convert the transformed scenarios into scenarios of un-transformed wind speeds by reversing the transformation performed in Sect. 3.2. A conversion from wind speed to power

scenarios is achieved via the International Electrotechnical Commission (IEC) turbine power curve for Class 2 turbines (IEC, 2007) before ramps are identified.

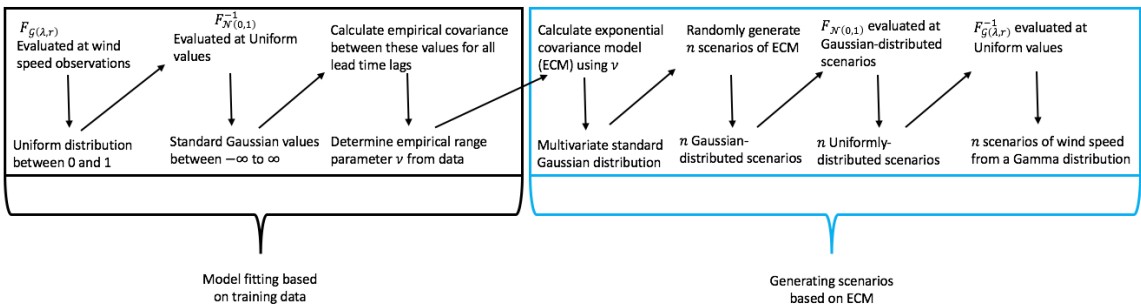

**Figure 4 Flow diagram of how the Gaussian copula method is used to generate $n$ number of wind speed scenarios from a multivariate Gamma distribution. Downward-pointing arrows show the result of the top processes. The diagonal-pointing arrows illustrate the next step in the method. The original wind speeds do not have to be transformed, but for this study, we begin this method using transformed wind speeds from a univariate Gamma distribution and result in $n$ number of scenarios of transformed wind speeds from a multivariate Gamma distribution.**

**3.3.2 Standard Schaake Shuffle**

We also generate forecast scenarios of transformed wind speed using the Schaake Shuffle method, which uses historical wind speed scenarios to determine serial dependence of the wind speed forecasts across forecast lead times. This method for generating multivariate forecasts is used widely for precipitation and temperature forecasts (Clark et al., 2004), but has not yet been applied for wind speed and power forecasts. This method generates wind speed forecast scenarios which can be

converted to power. Alternatively, the method could be used to generate power scenarios directly if given predictive distributions and observations of power. Forecast scenarios are easier to visualize in wind-speed-space (transformed wind speed for our data) because of the strong non-linearity of the power curve, so we discuss the method starting with predictive distributions and observations of transformed wind speed. For a given date, we construct 50 forecasts for each forecast lead time by breaking the predictive distributions in Sect 3.2 into 50 quantiles so that the $\eta$ forecasts are simply the $\eta$ quantiles of

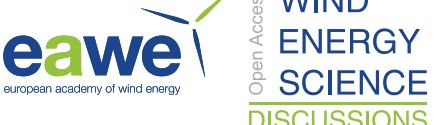



the predictive distribution. For 50 quantile forecasts, the quantile proportions range from 0.01 to 0.99 of the predictive distribution in increments of 0.02.

The next step in the Schaake Shuffle method is to select an identical number of observed historical scenarios of transformed wind speed. The historical scenarios are selected from the 50 available dates preceding the forecast initialization date, so that the historical scenarios of transformed wind speed are from a similar season. Alternatively, dates could be

pulled at random throughout the observed historical record. The method then ranks the 50 historical observations separately for each lead time and assigns the same ranking to the 50 sorted forecast quantiles (an illustration of this process for three historical scenarios and three forecast quantiles is shown in Figure 5b, c). The final step of the Schaake Shuffle method is to connect the ranked quantile forecasts across lead times to yield multivariate forecast scenarios (Figure 5d). For instance, a forecast quantile that is associated with historical scenario '3' at lead time 0 will connect to all forecast quantiles that are also

associated with historical scenarios '3' at their lead time (Figure 5d). This shuffling of forecast quantiles to match the rank of historical scenarios yields forecast scenarios that maintain a realistic temporal interdependence and shape across lead time while matching the predictive marginal distribution as described in Sect. 3.2.

Like in the Gaussian copula method, the generated scenarios of transformed wind speed forecasts from the Schaake Shuffle method can then be converted to power, if desired, to identify ramp events. Here, the selection of the historical

scenarios used in the Schaake Shuffle was ad hoc; the method does not make a preferential selection of dates. We next discuss two methods which preferentially choose historical scenarios that are most similar to the 1) quantiles of the forecast marginals and 2) the quantiles of the forecast marginals and also the quantiles of the wind speed difference between lead times. We distinguish between these three methods by referring to the standard method above as the Standard Schaake Shuffle (StSS), the first preferential method which will be discussed below as the Minimum Divergence Schaake Shuffle

(MDSS), and the enhanced preferential method also discussed below as the MDSS+.

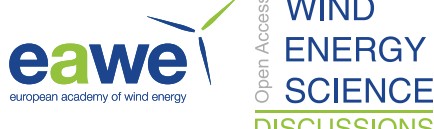

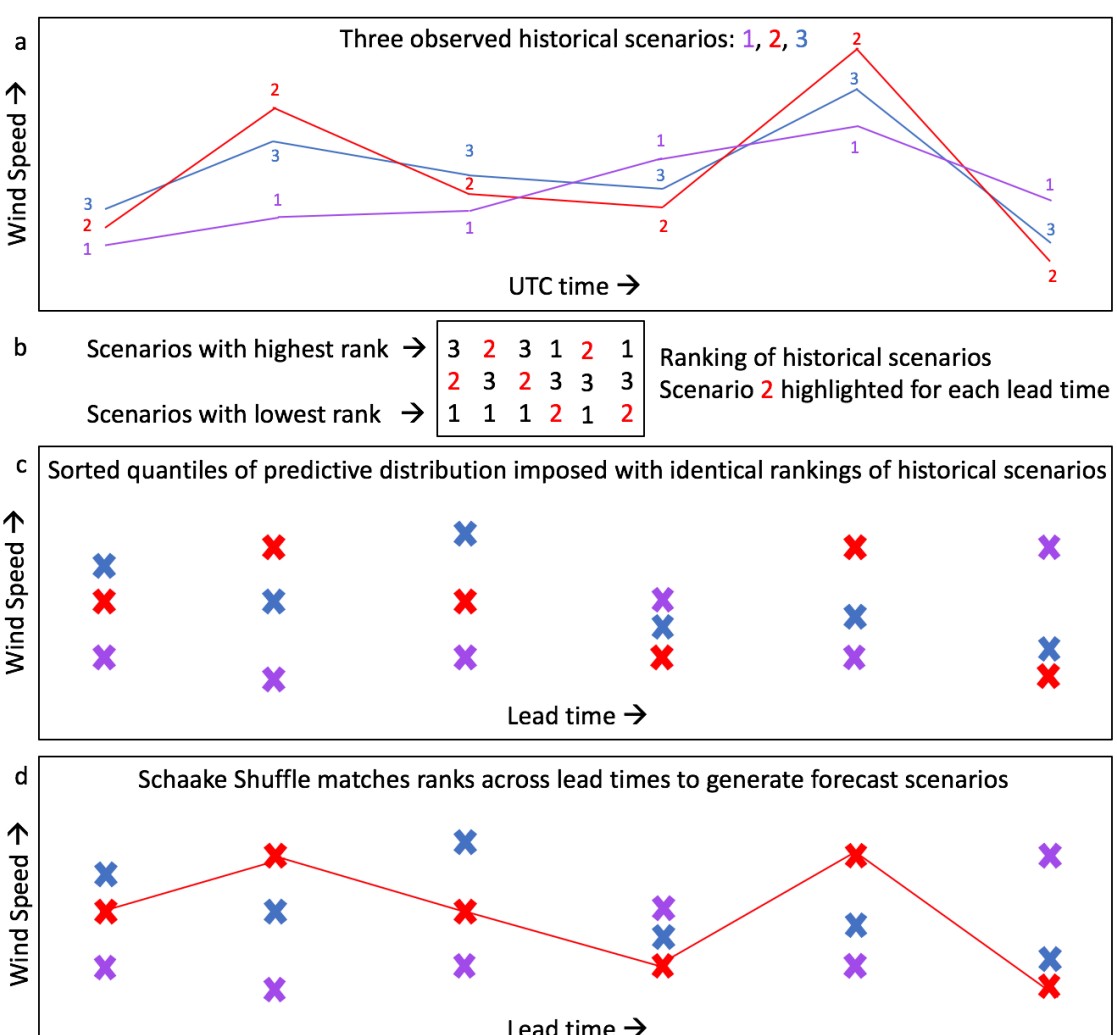

**Figure 5. Illustration of how the Schaake Shuffle method generates three wind speed forecast scenarios for a given date. For a given forecast date, three observed historical scenarios of wind speed are selected from the historical record (a). The historical scenarios are ranked (b) and then the same ranking is imposed onto the sorted forecast quantiles (c). The forecast quantiles are connected across forecast lead time according to their corresponding rank (d). In panel (b), we emphasize the ranking of the second historical scenario to show how the ranking of a historical event manifests in the shape of a forecast scenario in (d).**



### 3.3.3 Minimum Divergence Schaake Shuffle (MDSS)

The first of two methods that we use to preferentially select and generate probabilistic forecast scenarios of transformed wind speed is the Minimum Divergence Schaake Shuffle (MDSS) method (Scheuerer et al., 2017) . The MDSS follows the same procedures as the StSS method that impose the ranking of historical scenarios on sorted quantiles of the forecast distributions and that connect forecasted quantiles associated with one particular historical scenario across all lead times. Like for the StSS, the MDSS can also utilize historical observations from dates when no forecasts are available, an advantage over another variant of the Schaake Shuffle method introduced in Schefzik (2016).  The identical processes of the StSS and MDSS methods are shown in Figure 5. The MDSS deviates from the StSS in its selection of historical scenarios; the MDSS preferentially chooses dates such that the marginal distributions of the sampled historical scenarios are most similar to the quantiles of the post-processed forecast marginal distributions across all forecast lead times rather than a random or user-assigned selection of dates used for the StSS method. In the hydrological context discussed by Scheuerer et al. (2017), this preferential selection helped preserve features in the historical scenarios during the shuffling procedure shown in Figure 5c and d, and lead to improved multivariate probabilistic forecasts compared to StSS.

Because historical scenarios selected for the MDSS method are not limited to the most recent $\eta$ scenarios from the forecast initialization date as with the StSS method, the number of scenarios must be narrowed down to $\eta$ scenarios starting from the total number of candidate scenarios $N_0$ in the historical record, which for our dataset is $\approx 300 - 400$ scenarios for each initialization time and tower location. This selection seeks the $\eta$ historical scenarios that yield the least divergence[1] $\Delta_k^{\mathcal{H}}$ between the CDF of the forecast marginal distribution $F_k^f$ at each lead time $k$, and the empirical CDF $F_k^{\mathcal{H}}$ calculated from a set $\mathcal{H}$ of historical observation scenarios,

$$\Delta_k^{\mathcal{H}} = \int \left( F_k^{\mathcal{H}}(x) - F_k^f(x) \right)^2 dx. \tag{4}$$

Each scenario within the set $\mathcal{H}$ is evaluated for final selection based on whether the scenario results in a larger or smaller total divergence $\Delta_{tot}^{\mathcal{H}} = \sum_k \Delta_k^{\mathcal{H}}$ when it is removed from the calculation. If the scenario results in a smaller divergence when it is left out of the computation, then it is not an optimal choice. Conversely, if leaving out the scenario results in a larger divergence, then we know that the scenario is important for minimizing the divergence and should be kept as one of the final $\eta$ scenarios. Ideally, a set $\mathcal{H}$ that includes all possible candidate scenarios would be reduced to size $\eta$ one-by-one, but this is computationally expensive. Therefore, we use a sequence of $\mathcal{H}$ that reduces the starting number of candidate scenarios to test and eliminates more than one scenario with each iteration until $\eta$ scenarios are reached. For example, for the M5 tower

---

[1] Divergence in this study means the integral of the squared difference between two CDFs and is different from the divergence term $\nabla \cdot \boldsymbol{F}$ commonly used in meteorology, where $\nabla$ is the del operator and $\boldsymbol{F}$ is a meteorological field.



location at initialization time 00Z, there are $N_0 = 416$ total candidate historical scenarios, but we use the sequence 350, 300, 250, 200, 180, 150, 140, 130, 120, 100, 80, 70, 65, 60, 55, $\eta$, which reduces $N_0$ to $\eta = 50$ historical scenarios in 15 iterations rather than $N_0 - \eta$ iterations. Coding details of the method are given by Scheuerer et al. (2017) along with a computationally-efficient method for calculating the integral.

### 3.3.4 Enhanced version of the Minimum Divergence Schaake Shuffle (MDSS+)

Constraining the marginal distributions does not necessarily improve the representation of temporal gradients of the quantity of interest. If the HRRR forecasts of temporal wind speed changes have some skill, then using a predictive distribution of these differences explicitly in the MDSS algorithm might result in a better selection of historical dates that have similar temporal gradients. This formulation is the idea behind the final method we use to generate transformed wind speed scenarios. The final method is much like MDSS, but includes an additional term to explicitly capture the variation in wind

speed between neighbouring forecast lead times. For this enhanced MDSS method, $\eta$ historical scenarios are chosen that yield the least divergence from both the forecast marginal distributions and the forecast distribution of the lag1-h lead time differences of transformed wind speed. Forecast distributions of lag 1-h lead time differences are attained in the same way as forecast marginal distributions (Sect. 3.2), except that now we perform a regression on lag 1-h difference of transformed wind speed. Based on PIT histograms (not shown), the best predictive distribution that represents these differences for both

tower locations is the (non-truncated) logistic distribution. For this method, the $\eta$ historical scenarios that yield the smallest divergence when considering both the forecast marginal distributions and the forecast distributions of wind-speed differences are selected. To emphasize the temporal gradient between two neighbouring lead times, we assign more weight to the divergence term associated with wind speed differences. In this study, we weight the wind speed difference term as five times greater than the marginal distribution term. This method requires that the lag 1-h difference between lead times in the

historical scenarios best-match the lag 1-h differences of the forecast and is therefore an enhanced method to the MDSS.

### 3.3.5 Differences between historical observations selected by StSS, MDSS, and MDSS+

Marginal distributions of transformed wind speed of the historical scenarios used for each of the three Schaake shuffle methods (Figure 6a) and the distributions of the lag 1-h differences of those scenarios (Figure 6b) reveal that the MDSS and MDSS+ produce historical scenarios closer to the forecasted distributions than does the StSS method. Of course, the MDSS+

is the only multivariate method that utilizes the lag 1-h differences when selecting historical scenarios and for that reason, we see that the MDSS+ distributions for the lag 1-h differences (green boxes in Figure 6b) are often a slightly better match to the forecasted distribution (grey boxes in Figure 6b) than the regular MDSS or StSS methods (pink and blue boxes in Figure 6b, respectively). The MDSS+ method sometimes makes compromises in the selection of optimal scenarios for one of its two terms, because it seeks to find the historical scenarios that are an overall best match when considering both the quantiles



of the forecasted transformed wind speed distribution and the distribution of lag 1-hr differences of those wind speeds. Also, the MDSS and MDSS+ methods only have a limited set of historical dates from which they can choose scenarios, so we cannot expect a perfect match. Boxplots of the distributions of transformed wind speeds and the lag-1h differences of those wind speeds are not shown in Figure 6 for the GC method, because we wanted to point out the differences among the historical scenarios selected by each Schaake Shuffle method; the GC method does not use historical scenarios. Once the

historical scenarios are chosen, the quantiles of the forecast marginal distributions are reordered to have the same ranking of the corresponding historical scenarios. Like for the Gaussian copula and StSS methods, both the MDSS and MDSS+ scenarios are then transformed back into wind-speed-space and converted to power before identifying ramp events.

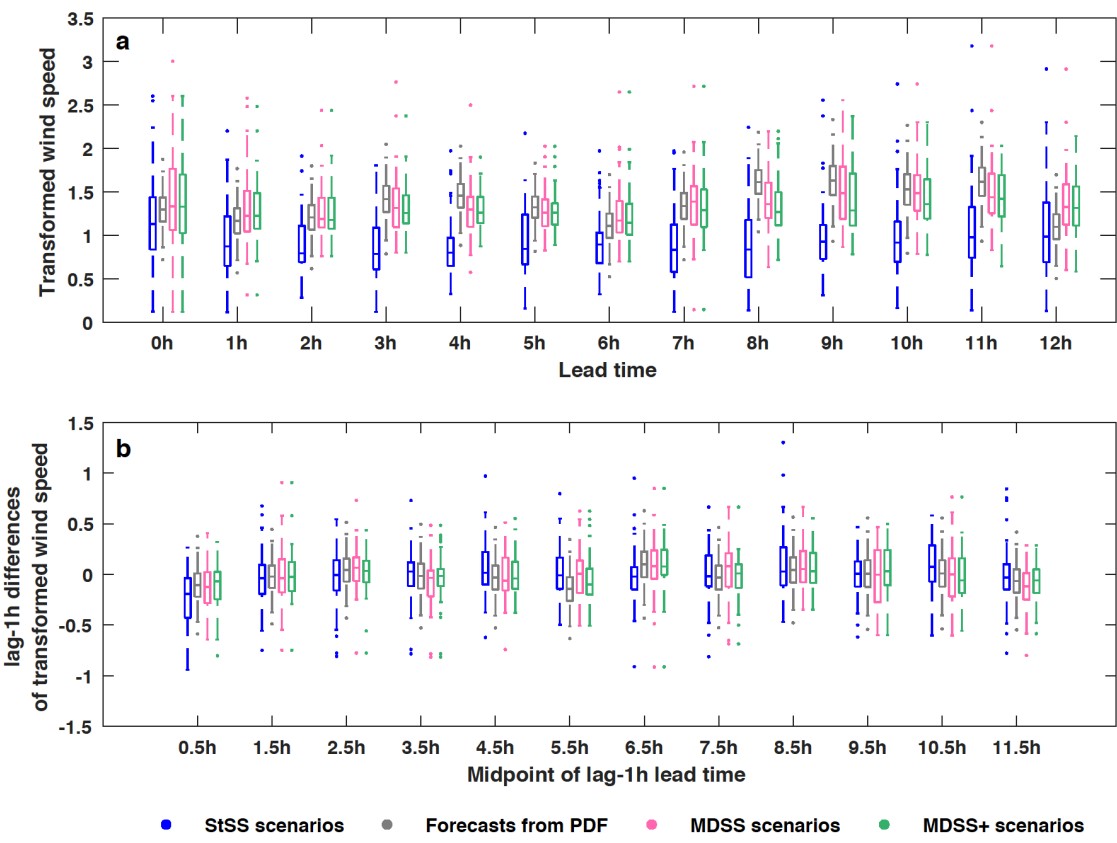

**Figure 6. Box plots of a) the forecast marginal distributions of transformed wind speed (grey boxes) and b) forecast lag 1-hr wind**
**speed differences (grey boxes) for 00Z forecast initialization on 28 March, 2015 at the PNW tower location. Distributions of transformed wind speed and lag 1-hr differences in transformed wind speed from the 50 historical scenarios used by the StSS (blue boxes) and selected by the MDSS (pink boxes) and MDSS+ (green) methods are also shown. The box plots display the**



interquartile range (rectangle region), the median (middle line within rectangle), outliers (dots), and values outside the
interquartile range but not considered an outlier (lower and upper whisker) of the distributions. Outliers are values more than 1.5
times the interquartile range.

## 4 Results

### 4.1 Verification of deterministic HRRR forecasts with observations

To provide a reference for the performance of predicting up- and down-ramp events, we first illustrate how ramps identified
from the raw HRRR forecasts compare to those identified from the observations at the M5 and PNW tower locations. The
correlation between ramps identified with the HRRR forecasts and observations are low at both tower locations (Figure 7)
ranging between 0.23 and 0.37. The ramp definition used for Figure 7 is different from the ramp definition discussed in Sect.
3.1, because it shows ramps identified with wind speed instead of power. This ramp definition is only used in Figure 7 to
show the magnitude of the change in wind speed that is observed and forecasted at each tower location during a period of
three hours. Utilizing the magnitude directly - rather than a particular exceedance event - eliminates the need to set any
particular threshold for a change in wind speed, which would be difficult to define anyway because of the nonlinear
relationship between wind speed and the power curve. The purpose of Figure 7 is to reveal biases in the HRRR forecasts and
differences between the two tower sites, while the analysis of power ramp events in the subsequent sections is more
applicable for decision-making of power grid operations.

At the M5 tower site, the HRRR predicts stronger wind speed ramps compared to observations; forecasted wind
speed ramps $\geq 5$ m s$^{-1}$ make up 40% of the total number of up- and down-ramps while the observed ramps of the same
magnitude only make up 33% of the total number of ramps. The HRRR generally under predicts the magnitude of wind
speed ramps at the PNW site; observed wind speed ramps $\geq 5$ m s$^{-1}$ make up 18% of the total number of up- and down-
ramps while the forecasted ramps $\geq 5$ m s$^{-1}$ only contribute to 9% of the total number of ramps. These percentages also
highlight that the M5 location has a greater percentage of observed ramps of the same magnitude than at the PNW location
(33% vs 18%), suggesting that the wind speeds at the M5 site are more variable than at the PNW site. The M5 tower is
located in a region of very complex terrain about 5 km east of the Colorado Front Range, which because of the atmosphere's
interaction with the mountainous terrain can cause large changes in wind speed over short periods of time. The PNW tower
is also located in a region of complex terrain near the Columbia River Gorge, but the terrain is not as complex as the M5 site.

The low correlation coefficients between the HRRR forecasts and observed wind speed ramps suggest that there is
some skill in the HRRR forecasts at predicting ramps, but the skill is limited and differs between up- and down- ramps. Low
correlation limits the extent to which statistical post-processing can improve the forecast. On the other hand, we have shown
that systematic over- and under-forecasting biases can be reduced with statistical post-processing. Moreover, the multivariate
methods discussed in this paper can provide information about the uncertainty of the forecast via generation of many
possible wind speed scenarios.

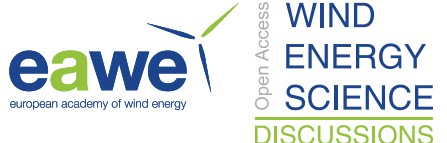


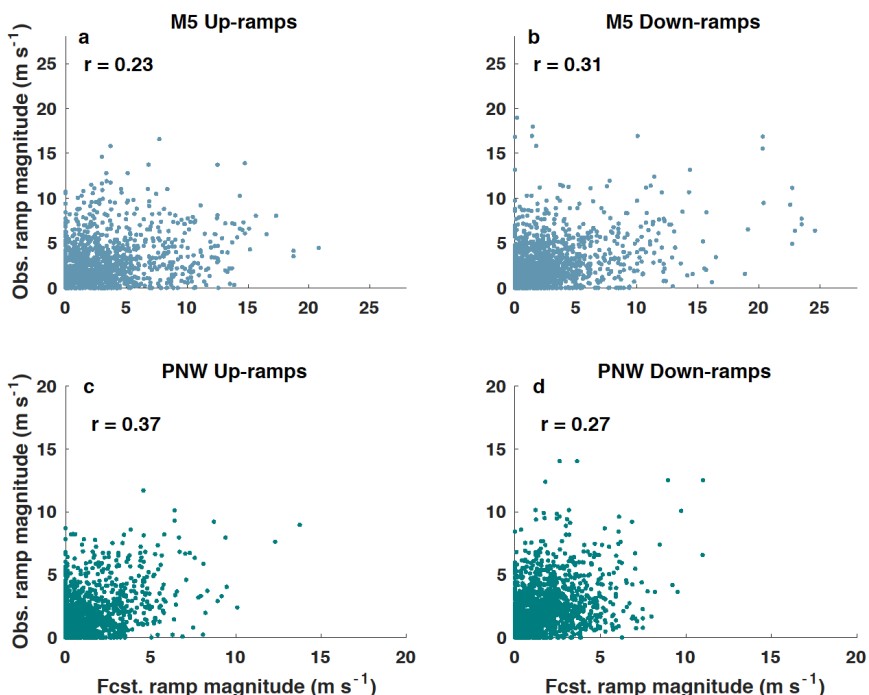

**Figure 7.** Observed (Obs.) and HRRR-forecasted (Fcst.) wind speed ramps during sliding 3-h time windows over a period of 12
hours. Absolute values of the ramp magnitudes are shown. Wind speed ramps are plotted for every sliding 3-h time window
starting at 00Z and for both the M5 (a, b) and PNW (c, d) tower locations. The correlation coefficient, r is displayed for each tower
location and type of ramp.

**4.2 Verification of multivariate methods compared with HRRR forecasts**

We now compare the various multivariate methods used to generate scenarios of transformed wind speed. To also compare
the different methods to the deterministic raw HRRR forecasts, we first employ an event-based metric to assess systematic
biases with regard to the frequency of ramp events. This metric counts the number of power ramps defined as in Sect. 3.1
identified from the scenarios generated by each method described in Sect. 3.3. The relative frequency of power ramps that
exceed $\xi = 60\%$ change in power capacity during 6 h for all days when forecasts and observations are available (Figure 8)
represents a climatology of up- and down-ramps for each tower location. The number of ramps identified in each 6-h



window of time for each of the 50 scenarios (1000 scenarios for Gaussian copula method) were averaged together and plotted as a single line in Figure 8. We again see a general over-forecasting bias of the number of ramp events (this time power ramps) produced by the raw HRRR forecasts compared to observations at the M5 tower (Figure 8c, d) and the opposite behaviour of the HRRR forecasts at the PNW tower (Figure 8a, b). The HRRR forecasts especially struggled with the diurnal cycle and magnitude of the relative frequency of up- and down-ramps at the PNW location. The HRRR predicted

the most up-ramps in the first four ramp windows (between 00Z and 9Z) and then levelled out for the remainder of the early morning while the observations show a minimum in up-ramps during the first four ramp windows and a maximum during the remaining windows, which suggests that the HRRR incorrectly captured the diurnal cycle. For the down-ramps, the HRRR forecasted a gradual increase in ramp events across all ramp windows, while the observations show a peak in down-ramps around the fourth ramp window ($\approx$ 9Z) followed by a gradual decrease in down-ramp events during the remainder of

the morning.

        The method that most-closely follows the ramp climatology of the observations (black line in Figure 8) is the StSS method followed by the MDSS+, MDSS, and lastly the Gaussian copula method. The StSS method has an overall better prediction of up- and down-ramp climatology than the raw HRRR forecasts when compared to a climatology of observed ramp events. As discussed in Sect. 3.3, the MDSS and MDSS+ methods make a preferential selection of historical scenarios

that minimize the divergence between the post-processed forecast and past scenarios and should yield scenarios more similar to the current forecast than the random or assigned scenarios used in the StSS method. Despite this preferential selection, the MDSS and MDSS+ methods do not outperform the StSS method in predicting the climatology of relative frequency of ramp events for this dataset. The reasons for this result are presented in the discussion for Figure 12. Before discussing reasons for why the more complex methods do not outperform the standard Schaake Shuffle method at predicting a climatology of ramp

events, we next examine a metric used to compare the skill of various probabilistic forecast methods to determine the differences in performance between the StSS and two MDSS methods.




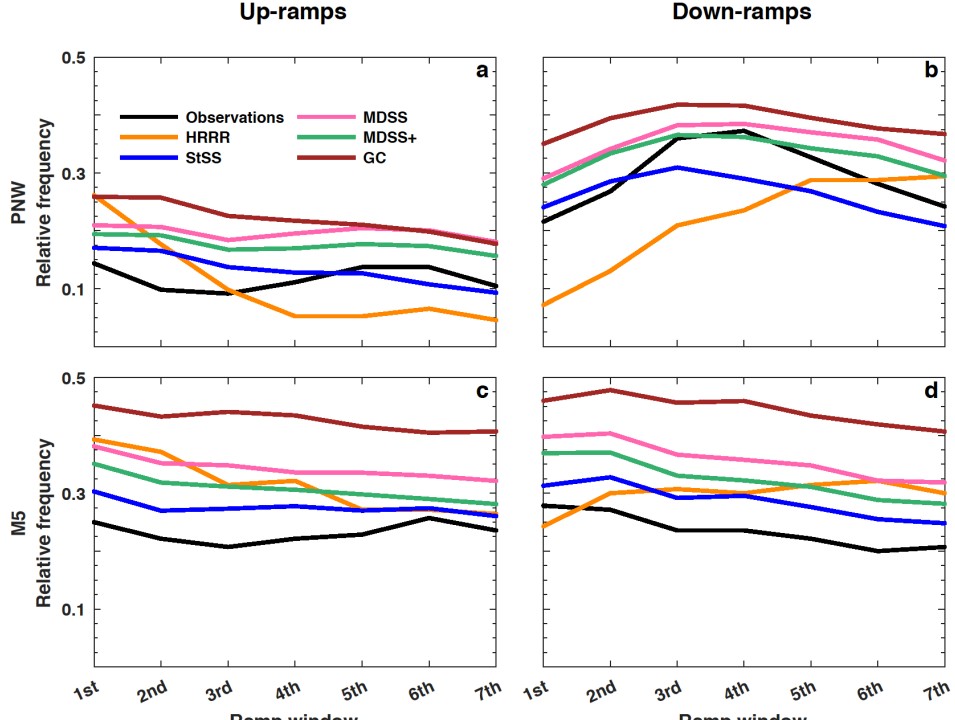

**Figure 8 Relative frequency of up-ramps (a, c) and down-ramps (b, d) identified with ≥ 60% change in power capacity during each 6-h time period (i.e., ramp windows) starting at 00Z. Relative frequencies of observed ramps (black lines) and forecasted ramps from the HRRR model (orange lines) are shown for the PNW (a, b) and M5 (c, d) tower locations. Relative frequencies are also shown for the four different multivariate methods: standard Schaake Shuffle (StSS, blue lines), Minimum Divergence Schaake Shuffle (MDSS, pink lines), enhanced Minimum Divergence Schaake Shuffle (MDSS+, green lines), and Gaussian Copula (GC, brown lines).**

The StSS, MDSS, MDSS+, and the Gaussian copula methods produce probabilistic forecasts of ramp events. To verify the skill of and compare among the different probabilistic methods, we compute Brier Skill scores (BSS). The Brier skill score quantifies the extent to which a forecast method improves the prediction of a two-categorical event compared to a reference forecast,

$$BSS = -\frac{BS_{fcst} - BS_{ref}}{BS_{ref}}, \qquad (5)$$



where $BS_{fcst}$ is the Brier score of the forecast and $BS_{ref}$ is the Brier score of the reference forecast. The Brier score is a strictly proper score that summarizes the accuracy of a probabilistic forecast; it is defined as the squared error of the

probability forecast of an event and the observed binary outcome (1 if the event happened, 0 if not). The events here are characterized by the exceedance of a particular ramp threshold $\xi$ during a ramp window size $h$. Climatological probabilities of occurrence of up- and down-ramp events with a particular $\xi$ and $h$ are used as the reference forecast. Before calculating the $BS_{fcst}$, we took the average of the binary event forecasts from all 50 scenarios (1000 scenarios for Gaussian copula method) for each method to create a probabilistic forecast with a value between 0 (no ramps occurred in any of the scenarios)

and 1 (ramps occurred in all of the scenarios). Brier scores were calculated for each type of ramp (i.e., up- and down-ramps with $\xi$ = 0.20, 0.40, 0.60, and 0.80 and $h$ = 3 h and 6 h). To quantify the sampling variability of the BSS induced by the limited data sample size, we first generated 100 bootstrap samples with replacement of the daily $BS_{fcst}$ and $BS_{ref}$ separately for each forecast initialization time (i.e., 00Z and 12Z). Then, we summed the 100 BS from each initialization time together before calculating the BSS to reduce sampling variability.

520         Box plots of the BSS for both tower locations and different types of power ramps reveal dependencies of the forecast skill on $\xi$, $h$, and tower location (Figure 9 and Figure 10). The most noticeable difference among the BSS is that the skill is generally higher for forecasts made at the PNW tower location compared to those at the M5 tower location for all types of ramps. Recall from Sect. 4.1 that the observed up- and down-ramps and those predicted by the HRRR had low correlation coefficients, which is why it is difficult to get positive skill with any of the methods at the M5 site; statistical

post-processing can correct for systematic forecasting biases, but it cannot improve random errors which lead to low correlation. Conversely, at the PNW site, there are overall higher correlation values (Figure 7) compared to those at the M5 site meaning that statistical post-processing will be more consequential. This behaviour results in generally positive and higher BSS for the PNW site than for the M5 site (Figure 9 and Figure 10). Greater positive skill is gained when we identify ramps in a window size of 6 h (Figure 10) instead of 3 h (Figure 9) for both the PNW and M5 sites, because timing errors are

less consequential when the time window is larger.

The multivariate methods do not present as much skill in forecasting down-ramps as they do in forecasting up-ramps at the PNW site, except for events with small (20%) power changes during 3 h. In Figure 7, the correlation between observed ramps and ramps forecasted by the HRRR is greater for up-ramps (0.37) than down-ramps (0.27) at the PNW site. At the M5 site, the correlation between observed and HRRR-forecasted down-ramps (0.31) is larger than for up-ramps

(0.23). We also note greater skill for the multivariate methods at predicting down-ramps opposed to up-ramps at the M5 site, which combined with the relative skill of up- and down-ramps at the PNW site suggests that the quality of the initial raw forecast skill impacts the amount of skill that can be gained from the probabilistic approaches.

How does skill vary among the different multivariate methods? The scenarios produced with the Gaussian copula method result in significantly less skill than all of the other methods for the M5 tower location and marginally less skill than



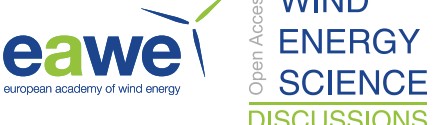

the other methods at the PNW site. The Gaussian copula utilizes an exponential covariance model that defines the temporal dependence of scenarios through the range parameter $v$, which was set to 2.5 and 1.5 for the PNW site and M5 site, respectively. Those parameters were selected based on empirical correlations, but did not yield the highest BSS. Instead, we achieved the highest BSS using $v = 3.5$ and 2.5 (not shown) for the PNW and M5 site, respectively, but these values are not apparent choices based on what the data suggest. The fact that a larger parameter produces a better result implies that a

generic exponential model may not be a good fit for these complex sites; the inclusion of local forecasts to create probabilistic scenarios is necessary to understand the correlation among lead times.

The most surprising result from the analyses is that the StSS method is more competitive than the MDSS method (and slightly more competitive than the MDSS+ method) despite the preferential selection of historical scenarios made by the MDSS method. The MDSS method selected historical scenarios of transformed wind speed that were most compatible

with the marginal distributions of the forecast day and theoretically should provide higher BSS than the StSS method, which only could use scenarios from the 50 available historical dates prior to the forecast day. The original MDSS method used by Scheuerer et al. (2017) worked well for precipitation events, but does not focus on the selection of historical scenarios based on their compatibility with forecasted (temporal) gradients, which are crucial for the prediction of ramps. This understanding led us to include an additional term in the MDSS method that is based on lag 1-h differences of transformed wind speed. The

modified MDSS method MDSS+, matches historical scenarios to not only the forecast marginal distributions but also to the forecast distributions of lag 1-h differences. The term of lag 1-h differences ensures a better selection of historical scenarios with ramps of similar slope or magnitude to the forecast.





**Figure 9 Box plots of Brier skill score (BSS) of power ramp events identified in 3-h time windows from forecast scenarios generated with the StSS (blue), MDSS (pink), MDSS+ (green), and Gaussian copula (GC, brown) methods. BSS are shown for the PNW tower location (a, c, and e) and the M5 tower location (b, d, f) and for up-ramps (filled boxplots) and down-ramps (non-filled boxplots). Ramp events with a power threshold of $\xi = 20\%$ (a and b), 40% (c and d), and 60% (e and f) change of the turbine power capacity are shown. The box plots display the interquartile range (rectangle region), the median (middle line within**
**rectangle), outliers (dots), and values outside the interquartile range but not considered an outlier (lower and upper whisker) of the distributions. Outliers are values more than 1.5 times the interquartile range. For reference, a line (black-dashed) showing zero skill is plotted.**








**Figure 10 Box plots of Brier skill score (BSS) of power ramp events identified in 6-h time windows from forecast scenarios generated with the StSS (blue), MDSS (pink), MDSS+ (green), and Gaussian copula (GC, brown) methods. BSS are shown for the PNW tower location (a, c, and e) and the M5 tower location (b, d, f) and for up-ramps (filled boxplots) and down-ramps (non-filled boxplots). Ramp events with a power threshold of $\xi$ = 40% (a and b), 60% (c and d), and 80% (e and f) change of the turbine power capacity are shown. The box plots display the interquartile range (rectangle region), the median (middle line within rectangle), outliers (dots), and values outside the interquartile range but not considered an outlier (lower and upper whisker) of the distributions. Outliers are values more than 1.5 times the interquartile range. For reference, a line (black-dashed) showing zero skill is plotted.**

We see that the median BSS using the MDSS+ forecast scenarios are often higher than those of the MDSS method and more competitive with the StSS method for all ramp types (Figure 9 and Figure 10). However, minute difference between the three Schaake shuffle methods are unperceivable, because of the limited sample size which resulted in considerable overlap between the BSS boxplots. To highlight the differences between the three Schaake Shuffle methods, we generated 25 years of synthetic wind speed observations and forecasts (Appendix B). These synthetic data underwent the same univariate post-processing steps (Sect. 3.2) as the real data before applying the different Schaake shuffle methods. Box plots of BSS using the synthetic data present positive skill, show considerably less variability than the real data, and highlight the differences between the MDSS and MDSS+ methods (Figure 11). The inclusion of the lag 1-h differences in the MDSS+ is essential to achieve optimal and competitive skill from the method when compared to StSS. Recall that the lag 1-h differences are weighted five times more than the forecast marginal distributions in the MDSS+, meaning that the transformed-wind-speed gradient between lead times is even more important to match than the forecast marginal distributions. With this additional term, the MDSS+ is as competitive as the StSS method at choosing scenarios that lead to skilful ramp forecasts.

Why is the rather simplistic StSS method as good (even better in regards to ramp frequency biases, see Figure 8) as the more sophisticated MDSS+ and significantly better than the MDSS method? Some insight is gained by analysing the lag 1-h differences of wind speed forecasts generated by the three different Schaake Shuffle techniques. We compute absolute lag 1-h differences of observed wind speeds and those of the historical observed wind speed scenarios selected by the StSS, MDSS, and MDSS+ methods before shuffling. For each method, the absolute lag 1-h differences are calculated for each date and for each 12 pairs of lead times. For each date and paired lead time, the lag 1-h differences are then stratified according to the corresponding HRRR wind speed forecast. Lag 1-h differences from all dates and paired lead times associated with a certain range of HRRR forecasted wind speeds are then averaged together before plotting (Figure 12a). A dependency between the magnitudes of lag 1-h differences and HRRR wind speed forecasts emerge. The magnitude of the observed lag 1-h differences increases as the HRRR forecast wind speed increases, which suggests that higher wind speeds correspond to larger fluctuations in wind speed. Because the StSS method does not depend on the HRRR forecast to select historical scenarios, the lag 1-h differences of the StSS historical scenarios are independent of the magnitude of the HRRR forecast wind speed. This result is demonstrated by the relatively flat StSS (blue) curve in Figure 12a. Conversely, the MDSS and





MDSS+ methods make a preferential selection of past observations based on the current HRRR forecast wind speed. The result is that the MDSS and MDSS+ methods produce curves (pink and green lines, respectively in Figure 12a) of lag 1-h differences qualitatively similar to the observed curve (black line in Figure 12a).

This better initial selection of scenarios, however, is offset by the effects of the shuffling procedure. Panel b) in Figure 12 shows the mean absolute lag 1-h differences after shuffling. For the StSS, the shuffling makes the lag 1-h

differences more similar to the observed lag 1-h differences; lag 1-h differences decrease for low HRRR forecast wind speeds and increase for high HRRR forecast wind speeds during the shuffling procedure. For the MDSS and MDSS+ methods, shuffling always results in a slight increase of the magnitudes of lag 1-h differences (pink and green lines in Figure 12b). This increase after shuffling the scenarios explains why the MDSS and to a lesser extent, the MDSS+ have a tendency to over-forecast the magnitude and frequency of wind speed ramps (see Figure 8).

Lastly, we investigate why the shuffling procedure affects the historical StSS scenarios differently than the MDSS and MDSS+ scenarios. We conjecture that one of the reasons for this effect is the difference in spread of the scenarios used by each method before shuffling. We quantify the spread as the mean absolute difference between the historical scenarios. Since the historical StSS scenarios are chosen unconditionally, the spread of its marginal distribution (see blue line in Figure 12c) approximates the climatological spread of actual observed wind speeds. Preferential selection performed by MDSS and

MDSS+ significantly decreases the spread of the historical marginal distributions with the exception for high HRRR wind speed forecasts where the prediction uncertainty can exceed the climatological spread (i.e., exceed blue line). This initial reduction in spread, however, reduces a side effect entailed by StSS: the shuffling procedure squeezes together scenarios as the unconditional spread is transformed into a forecast-informed spread. By doing this, the shuffling procedure typically reduces the fluctuations present in the historical scenarios. Because all of the Schaake Shuffle methods discussed herein use

the same quantiles of a particular forecast distribution, all methods have the same spread after shuffling (gray line in Figure 12c). Since the MDSS and MDSS+ historical scenarios already have low spread, shuffling does not change their characteristics as much as it does for the StSS historical scenarios; the level of fluctuations is similar before and after shuffling. In other setups, the shuffling side effect can be unwanted, but in the present setup, it seems to benefit the StSS method and results in the overall most accurate level of wind speed fluctuations.






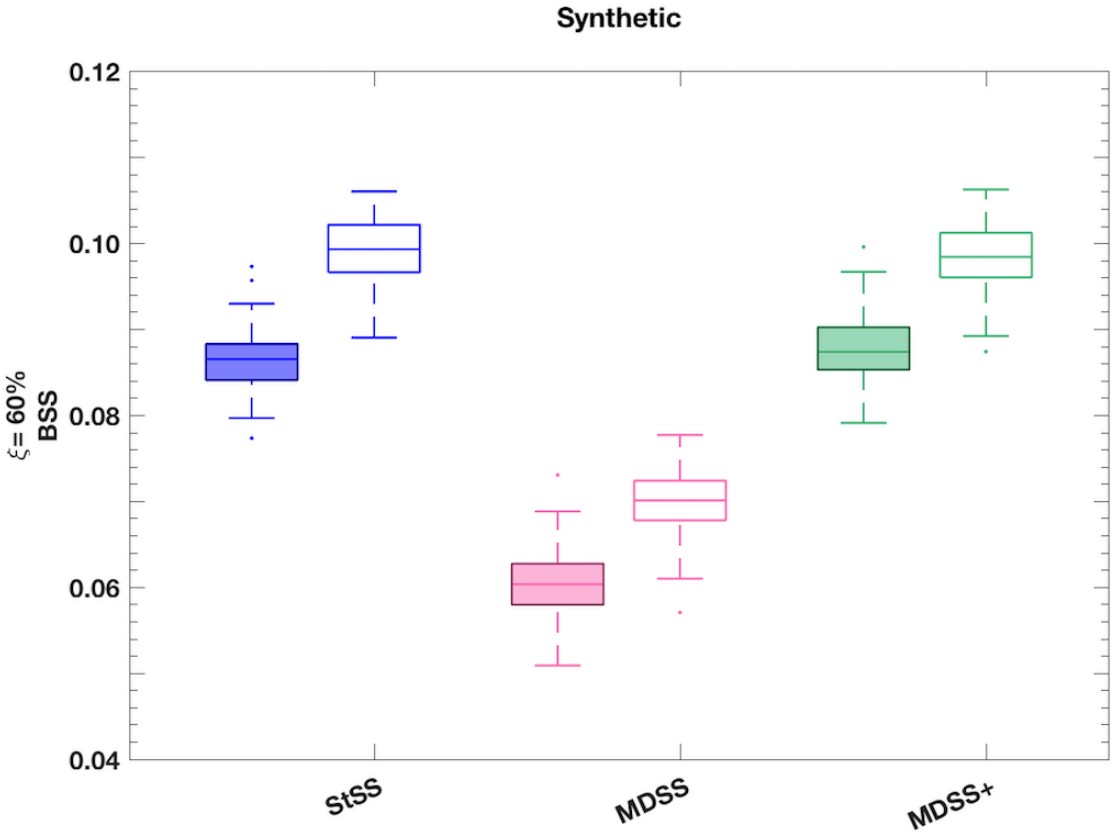

**Figure 11 Box plots of Brier skill score (BSS) of power ramp events identified in 6-h time windows from synthetic forecast scenarios generated with the StSS (blue), MDSS (pink), MDSS+ (green), and Gaussian copula (GC, brown) methods. BSS are shown for the PNW tower location for up-ramps (filled boxplots) and down-ramps (non-filled boxplots). Ramp events with a power threshold of $\xi = 60\%$ change of the turbine power capacity is shown. The box plots display the interquartile range (rectangle region), the median (middle line within rectangle), outliers (dots), and values outside the interquartile range but not considered an outlier (lower and upper whisker) of the distributions. Outliers are values more than 1.5 times the interquartile range. For reference, a line (black-dashed) showing zero skill is plotted.**




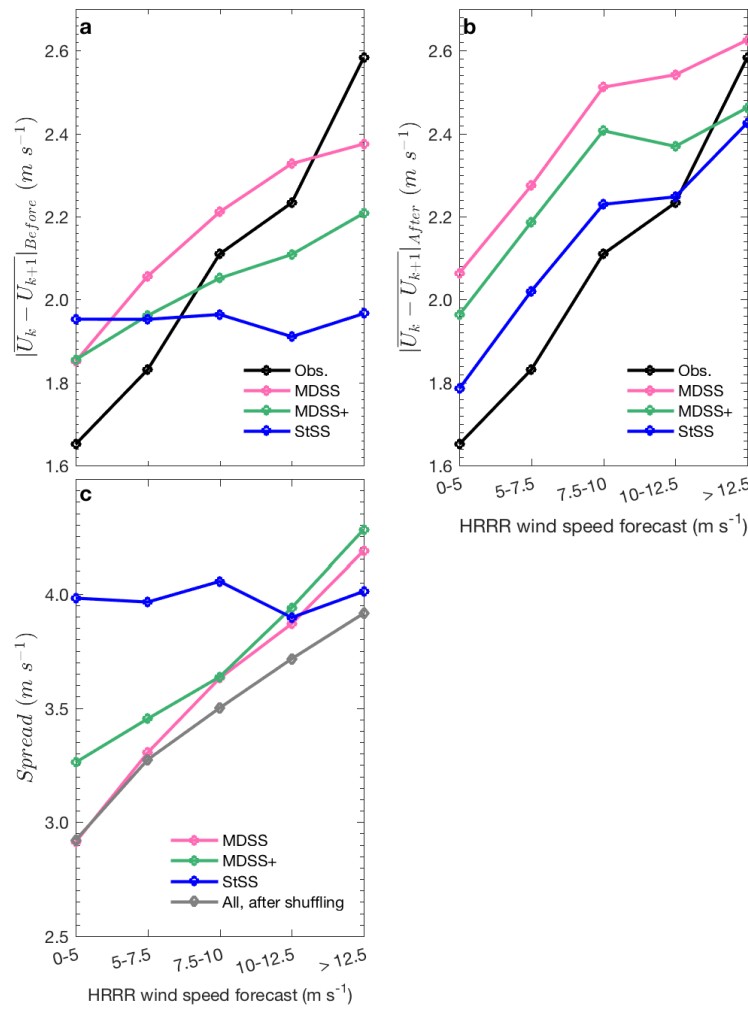


**Figure 12 Statistics of observed wind speeds ($U$), unshuffled historical observed wind speed scenarios used by each Schaake Shuffle method, and the shuffled wind speed forecast scenarios stratified according to the magnitude of the HRRR wind speed forecasts at the respective time. Mean absolute lag 1-h wind speed differences ($\overline{|U_k - U_{k+1}|}$) of observations and historical scenarios before (a) and after (b) shuffling, and the marginal spread of the unshuffled (pink, blue, and green lines) and shuffled**
**(gray) historical scenarios (c) are shown. The spread is quantified as the mean absolute difference between scenarios.**



## 5 Summary and conclusion

Wind power ramps present challenges to wind power forecasters and the electrical grid, because they cause sharp changes in power production in time periods of minutes to hours. Better forecasts of ramp events can lead to more reliable wind power generation and less strain on the power grid. Generally, wind farm operators rely on a single forecast of persistence to determine power fluctuations over the next 30 mins to an hour, which is not suitable during ramp events. Numerical weather prediction and statistical post-processing techniques can improve ramp forecasts by predicting rapid future fluctuations in wind speed and power and by providing uncertainty information to those forecasts.

In this paper, we used observed 80-m wind speeds from tall meteorological towers located in Boulder, Colorado (M5 site) and in eastern Oregon (PNW site). We also used forecasts of 80-m wind speeds from the HRRR model to create probabilistic forecasts of up- and down-ramp events. With these data, we presented how to obtain probabilistic wind speed forecasts by first correcting biases in the forecasts and then applying one of the four multivariate methods discussed to generate scenarios of wind speed. We used the IEC power curve to convert scenarios of wind speed to scenarios of power before identifying ramps. Alternatively, a relationship between measured wind speed and power output from a training dataset could be used to bypass the use of a power curve for future wind speeds (Lange and Focken, 2006). Employing stochastic power curves (Jeon and Taylor, 2012) would also take the conversion uncertainty into account. Because our study was focused on the evaluation and comparison of multivariate statistical post-processing methods and wind speed to power conversion affects all methods in the same way, using a fixed power curve warrants a fair comparison. If observed power production rather than observed wind speed was used as the 'ground truth', an inverse (power-to-wind speed) transformation could be employed to reconstruct the associated wind speeds (Messner et al., 2014), and the conversion uncertainty would be accounted for implicitly.

Before generating the scenarios, we removed the seasonal cycle and corrected for heteroscedasticity within the observations and raw HRRR forecasts by applying a power transformation. We then regressed the transformed observations on the transformed forecasts to obtain regression coefficients. The mean and standard deviation of marginal predictive distributions for each forecast initialization and lead time were determined by inserting future forecasts into the fitted regression model with these coefficients. We tested three candidate predictive distributions and found that the gamma distribution and the truncated logistic distributions were the best fits for the M5 and PNW tower locations in regards to wind speed, respectively. We determined that these predictive distribution models were suitable to represent observations based on uniform PIT histograms and low CPRS values. This approach to obtaining marginal predictive distributions is rather simple, but given the limited amount of data that remained after filtering, we thought that a stable parameter estimation for a more complex model was not warranted. A larger training dataset would allow one to account for forecast biases that vary with wind direction (Eide et al., 2017), or to use an analog-based regression approach similar to the method proposed by Junk et al., (2015), and include analog predictor variables related to atmospheric stability.



The marginal predictive distributions provided uncertainty information for each lead time, but did not inform us
680    about the interdependence structure across all lead times. To construct this interdependence, we first used the Gaussian
copula technique following Pinson and Girard (2012), which relates the predictive distributions across all lead times by
utilizing an exponential covariance model of Gaussian random variables. We used a random number generator to generate
1000 scenarios of wind speed using this method.  The Gaussian copula method is based on parametric assumptions that may
not be an adequate representation of the interdependence between observed wind speeds at different lead times, so we tested
three new methods of generating scenarios of transformed wind speeds. The standard Schaake Shuffle (StSS), the minimum
divergence Schaake Shuffle (MDSS), and the enhanced version of the MDSS (MDSS+) methods all use historical observed
scenarios to inform how marginal predictive distributions should be connected across all lead times, which results in more
realistic forecast scenarios.

The StSS method only used an ad hoc selection of historical scenarios while the MDSS and MDSS+ made
preferential selections of historical scenarios that best matched the forecast marginal distributions (MDSS) or matched both
the forecast marginal distributions and the forecast distributions of the lag 1-h differences of transformed wind speed
(MDSS+). Even with these modified version of the Schaake Shuffle, we found that the StSS method provided the highest
Brier skill scores overall using real data. However, all of these methods provided improvements over the raw HRRR
forecasts, which struggled to capture the diurnal cycle and magnitude of the relative frequency of up- and down-ramp events.
These methods also reduced the over- and under-forecasting biases of the raw forecasts at the M5 and PNW tower locations,
respectively. We compared the three Schaake Shuffle methods at forecasting ramp events using a dataset of 25 years of
synthetic forecasts and observations to emphasize the differences among the multivariate methods without constraints from
the limited real dataset. We found that the MDSS+ method had significantly higher skill compared to the MDSS and was
competitive with the StSS method suggesting that inclusion of the lag 1-h wind speed differences is a key component to
accurate forecasting of ramp events when preferentially selecting historical scenarios.

We were limited with how much improvement statistical post-processing could provide with the real data, because
the correlation between the observations and HRRR forecasts of up- and down-ramps were low. However, we still achieved
some positive skill by reducing over- and under-forecasting biases and by employing the multivariate methods to generate
probabilistic forecasts for the PNW tower which had overall higher correlation coefficients than that of the M5 tower
location. Generally, the greatest skill was achieved for the prediction of up-ramps at the PNW site for which also happened
to be the ramp type associated with the highest correlation.  This dependence on initial forecast skill is encouraging, because
it suggests that for sites with fewer random errors and better skill (e.g., sites over flat terrain), we may be able to achieve
significant improvement in forecast skill using these multivariate methods. A longer record of historical scenarios would also
be advantageous, because it would increase the likelihood that forecasts would have a good match with past events for
selection by the MDSS and MDSS+ methods.



We demonstrated how statistical post-processing can correct forecast biases of up- and down-ramp events and how multivariate statistical methods can be used to generate probabilistic forecasts of wind speed and power scenarios. These methods can be implemented for real-time wind-farm operations using historical observations at a particular wind farm to gain uncertainty information regarding ramp forecasts. We used the generic IEC power curve to convert wind speed
scenarios to power scenarios, but wind power forecasters should use their own turbine-specific power curves to further reduce uncertainty. Additionally, these methods are applicable with other numerical weather prediction models besides the HRRR model. Therefore, wind power forecasters can use forecasts from their proprietary models as long as observations are available during the same time period for verification. Enhancements to the forecasts provided by gaining uncertainty information should help with decision-making in the energy-sector not only for direct power generation, but also for
scheduling the availability of transmission lines, energy reserves, and energy trading. Future research that could improve these methods includes improvement to raw forecasts via various methods (e.g., increased grid resolution and improved physics parameterizations), using additional predictors in the regression analysis of the univariate data (e.g., temperature and wind direction), and performing these methods for sites that generally yield higher quality forecasts. Overall these methods may find utility in assessing risks of other wind-speed dependent phenomena like wildfire propagation or pollution
dispersion.

**Data Availability**

Data from the M5 meteorological tower are available at https://wind.nrel.gov/MetData/135mData/M5Twr/. Data from the PNW tower are available by request from Avangrid (contact Michael Zulauf, Ph.D). The HRRR forecast data are available
from the National Oceanic and Atmospheric Administration Global Systems Division (contact Eric James or Stanley Benjamin).

**Acknowledgements**

The authors express appreciation for Eric James and Stanley Benjamin at NOAA and CIRES for providing the raw HRRR data and NREL for providing the M5 tower data. The authors express great appreciation to Avangrid for the collection and
provision of the data from the PNW meteorological tower. We are greatly appreciative of Laura Bianco and Irina Djalalova from NOAA and CIRES and James Wilczak from NOAA for sharing their expertise of wind ramps and of the PNW tower location. Funding for this work was provided by NOAA under the federal Pathways Program.




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

## Appendix A: Determination of range parameter for exponential covariance model

The range parameter $v$ defines the decay of correlation of the exponential covariance model (Eq. 3) among lead times. We

determined the range parameters for use in the Gaussian copula method (Sect. 3.3.1) by choosing values of $v$ that best-

aligned with the observed correlation of the Gaussian wind speeds at each tower location and forecast initialization time. We

used $v = 2.5$ (associated with the purple lines in Figure A1a and b) and $v = 1.5$ (associated with the orange lines in Figure

A1c and d) for the PNW and M5 tower locations, respectively. We only considered lagged-lead time correlations out to 6 h,

because our largest ramp window size is 6 h.

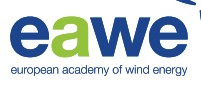
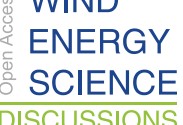


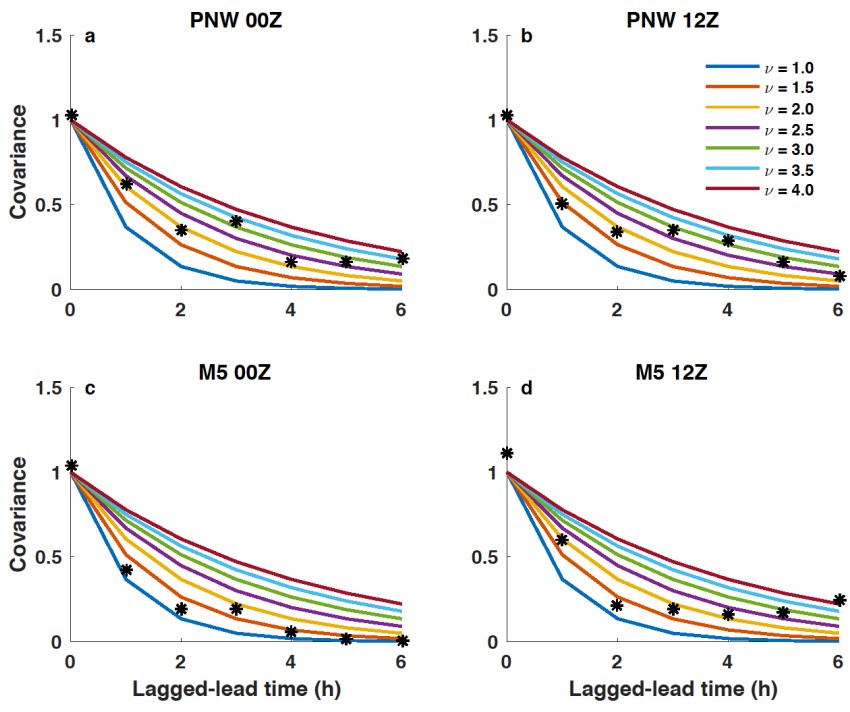


**Figure A1.** Exponential covariance models (solid lines) given different values of the range parameter $\nu$ ranging from 1 to 4 in increments of 0.5 as a function of lagged-lead time. Overlaid are the observed covariance values (black asterisk) of the Gaussian wind speed values. Covariance is shown from the PNW (a, b) and M5 (c, d) tower locations for 00Z (a, c) and 12Z (b, d) forecast initialization times.

**Appendix B: Generation of a synthetic dataset to overcome sample size limitations**

To generate a synthetic wind speed dataset (deterministic forecasts and observations) we use again a Gaussian copula approach, now applied to unconditional (climatological) marginal distributions. For simplicity, we assume the same climatology at each day of the year and each time of the day. Serial dependence in the Gaussian space is modelled via AR(1) processes, i.e. autoregressive processes of order one, that are used to generate two dependent time series $(z_t^{(x)})_{t=1,..,T}$ and

$(z_t^{(y)})_{t=1,..,T}$ with time index ranging from 1 to T. We proceed in two steps:

1. Simulate a bivariate Gaussian time series with zero mean and marginal variances equal to 1
   - let $\rho = 0.8$ be the correlation between the forecast and observation time series

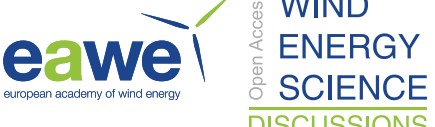



- simulate an AR(1) time series $(z_t^{(y)})_{t=1,...,T}$ corresponding to the 25 years of data, using $\phi = e^{-0.5}$ as the autoregression
  parameter and $\sigma^2 = 1 - \phi^2$ as the variance of the driving white noise process
- simulate another AR(1) time series $(\varepsilon_t)_{t=1,...,T}$ with the exact same specifications
- define a third time series $(z_t^{(x)})_{t=1,...,T}$ as $z_t^{(x)} = \rho \cdot z_t^{(y)} + \sqrt{1-\rho^2} \cdot \varepsilon_t$

By this construction, the correlation coefficient of the time series $(z_t^{(x)})_{t=1,...,T}$ and $(z_t^{(y)})_{t=1,...,T}$ at each time t is $\rho$.

2. Transform to a bivariate time series with gamma-distributed margins
- denote by $F^{-1}_{G(3,3)}$ the inverse CDF of a gamma distribution with shape parameter 3 and scale parameter 3
- denote by $\Phi$ the CDF of a standard Gaussian distribution
- the observation time series is then defined by $y_t = F^{-1}_{\Gamma(3,3)}(\Phi(z_t^{(y)}))$ $t = 1,...,T$
- the forecast time series is defined by $x_t = F^{-1}_{\Gamma(3,3)}(\Phi(z_t^{(x)}))$ $t = 1,...,T$
