# Peer review of "Generating wind power scenarios for probabilistic ramp event prediction using multivariate statistical post-processing"

_Wind Energy Science, 2018_

## Referee Comment (RC1) · J. W. Messner (Referee) · 14 Mar 2018

This paper presents and compares different multivariate approaches for probabilistic wind power forecasting to forecast ramp events. Accurate ramp forecasting is a very relevant topic in the wind power forecasting community and I think this manuscript is a valuable contribution to this topic. The manuscript is well written and I enjoyed reading it very much.

Even though, the data sets in the case studies are rather small, the results indicate advantages and disadvantages of the different methods, which are then investigated more closely in a simulation study. I like that the authors do not provide too many tech-

nical details, this facilitates the reading and I believe that it is still possible to reproduce most of the results.

There is not much to criticize and I only have a few minor comments that I listed below.

**More specific comments:**

1. The result section focuses mainly on the differences between the different Schaake shuffle methods. Although this is certainly interesting I think the differences between the Gaussian copula nad the Schaake shuffle are also very interesting and could be investigated and discussed a bit more. The Gaussian copula approach is probably better known in the wind power forecasting community and it would therefore be great to have a more extensive discussion on when and why this approach fails and Schaake shuffle approaches should be preferred.

2. In the definition of ramp events, it is not fully clear to me how $\Delta p_{max}$ and $\Delta p_{min}$ are derived. From Figure 1 it appears to be the sum of consecutive power differences that have the same sign. Related to that, if e.g., there is an up-ramp that is interrupted by a short down-ramp (e.g. 0,0.5,0.4,0.9), , is it then not classified as ramp?

3. The description of the fitting approach for the predictive distributions (Section 3.2) is also a bit unclear. As far as I understand, ordinary least squares regression is used to fit the transformed observations to the forecasts (and the forecasts to the seasonal cos and sin terms). Forecasts from this OLS model and its residual standard deviation are then used to derive parameters of truncated normal, truncated logistic or gamma distributions. How are the OLS outputs related to the distribution parameters? Did you simply use the method of moments? Somehow OLS assumes the residuals to follow a normal distribution and therefore this approach, if I understood it correctly, seems a bit weird. Why are you not using a distributional regression (i.e. EMOS like) approach where you directly fit the

distribution parameters as functions of the NWP forecasts and use a maximum likelihood or minimum CRPS framework?

**Minor comments:**

1. L70, why is it a disadvantage to use nonparametric methods.

2. Section 2.2: What are the lead times that are regarded. 1-12h? How long does it take the HRRR model to compute these forecasts and when do they become available? I.e., if the computation takes more than 1 hour the 1h forecast is not very useful.

3. Caption of Figure 1: I think a window size of h=4 not h=3h is shown

4. P9L219: Not only the red dots but also black dots show the annual cycle.

5. P12-13: In the description of the Gaussian copula approach, it should be clarified that the exponential covariance model (ECM) is an estimate of the covariances and that random number generation is based on a multivariate Gaussian distribution with these estimates as covariance parameter.

6. Section 3.3.3: I expect $\eta$ to be a quite important parameter and its optimal value clearly depends on the size of the data set. How did you select $\eta = 50$ and did you test the effect of this selection on the results?

7. P19,L451-452: "On the other hand, we have shown that systematic over- and under-forecasting biases can be reduced with statistical post-processing". I did not find any results showing this in the manuscript.

8. Last paragraph on page 24: I do not fully agree that StSS is more competitive than MDSS and MDSS+. The differences mostly do not seem to be significant and e.g., for down-ramps at PNW StSS actually performs worse.

9. Caption of Figure 11: There are no Gaussian copula results shown

10. Summary and discussion: It could be summarized here again, why multivariate methods are required to predict ramp events.

---

## Referee Comment (RC2) · E. Simon (Referee) · 22 Mar 2018

The research work utilizes operational HRRRv2 NWP data, along with four methods of statistical post processing to transform deterministic wind speed forecasts into hourly probabilistic up and down-ramp predictions over sliding windows for two sites with complex terrain. The article is well written and is presented clearly and concisely. Ramp forecasting is a very important field which can lead to significant reductions in wind power balancing costs, and I believe that this original research work makes a valuable contribution to the topic. Therefore I recommend to the editor that this article is accepted for publication. The scientific approach is clear and well thought out, and the

methods are presented in a reproducible and logical manner.

I have a few points that I ask the authors to consider for the final manuscript:

- The observations themselves also have underlying uncertainties. Has the instrumentation been recently calibrated? You can likely disregard this (but still good to mention) since the time averaging will reduce the statistical uncertainty (type A)

- How far are both sites from the HRRR model grid point? If the resolution is 3km, the closest point may not be representative (especially in complex terrain). Including a map with the mast locations and grid points would be a good addition.

- A comparison of these methods with a baseline (e.g. persistence or climatology) would strengthen the results

- Is there a reason to not first evaluate at a site with less complex flow, where the local observations are better correlated with model predictions? Especially since these methods have not been demonstrated before

- It isn't entirely clear, since you say that M5 obs are averaged to 10 minutes, while PWN are averaged to 1 minute. Is that only raw data and then they are both averaged again later to 1-hour?

- Do you expect any improvements with the HRRRv3 model changes?

- A note about processing time and ability to run these forecasts operationally would be useful information to provide

- Skill scores for the M5 site using synthetic data are not shown (Figure 11 only for PNW). Are the results similar?

[Figure]

- The caption on pg. 26 should be on the same page as the figure. Same with pg. 27 to reduce whitespace

- I agree that a continuation of the work should include wind direction and atmospheric stability. Especially since you have observations to properly classify these regimes. I look forward to reading future work from your group!

---

## Author Comment (AC1) · 16 Apr 2018

**Responses to reviewers, April 16, 2018**

Reviewer comments are typed in black.

Responses by Worsnop et al. are typed in purple.

We thank the referees for their comments, which have helped us clarify and improve the presentation of our work.

**Responses to reviewer, J. W. Messner**

This paper presents and compares different multivariate approaches for probabilistic wind power forecasting to forecast ramp events. Accurate ramp forecasting is a very relevant topic in the wind power forecasting community and I think this manuscript is a valuable contribution to this topic. The manuscript is well written and I enjoyed reading it very much.

Even though, the data sets in the case studies are rather small, the results indicate advantages and disadvantages of the different methods, which are then investigated more closely in a simulation study. I like that the authors do not provide too many technical details, this facilitates the reading and I believe that it is still possible to reproduce most of the results.

There is not much to criticize and I only have a few minor comments that I listed below.

**More specific comments:**

1. The result section focuses mainly on the differences between the different Schaake shuffle methods. Although this is certainly interesting I think the differences between the Gaussian copula and the Schaake shuffle are also very interesting and could be investigated and discussed a bit more. The Gaussian copula approach is probably better known in the wind power forecasting community and it would therefore be great to have a more extensive discussion on when and why this approach fails and Schaake shuffle approaches should be preferred.

We have added the following discussion:

L630: In fact, the average BSS from predicting up- and down-ramps using the Gaussian copula method is highly sensitive to the empirical range parameter used in the exponential covariance model (Table 2). For example, for the M5 location, $v$ = 4.5 yields the closest BSS values (Table 2) to those calculated for the Schaake Shuffle methods (Fig. 11) for all power thresholds and ramp types. This value of $v$ would also reduce the number of Gaussian copula ramp events that are currently over-forecasted in Fig. 9. However, based on the empirical covariances obtained for the M5 tower (Fig. A1), selecting a $v$ value this large did not seem plausible. It is possible that the assumption of an exponential covariance model (suggested by Pinson and Girard 2012) is not ideal for this setup, but with the limited training dataset, we felt that a parametric assumption was necessary to control sampling

variability of the estimated covariance matrix. Even then Fig. A1 suggests that stronger correlations than the empirical correlations would lead to better results. Our conclusion from results in Table 2 is that selecting an appropriate $\upsilon$ value before generating the Gaussian copula scenarios is critical but difficult to achieve with the usual statistical diagnostics. The Schakke Shuffle approaches do not rely on the selection of a sensitive parameter, which could make these Schaake Shuffle methods more preferable. Additionally, because the Gaussian copula method uses random sampling rather than quantile sampling, the Gaussian copula method requires many more scenarios to represent the distribution than do the Schaake Shuffle methods. From an operational perspective, too many scenarios (e.g., 1000 vs 50) may add unnecessary complication to the forecasting process.

2.  In the definition of ramp events, it is not fully clear to me how pmax and pmin are derived. From Figure 1 it appears to be the sum of consecutive power differences that have the same sign. Related to that, if e.g., there is an up-ramp that is interrupted by a short down-ramp (e.g. 0,0.5,0.4,0.9), is it then not classified as ramp?

$\Delta p_{max}$ and $\Delta p_{min}$ are defined as the largest pairwise difference in power in a given window of time (i.e., largest increase over time for $\Delta p_{max}$ and largest decrease over time for $\Delta p_{min}$). If the largest increase or largest decrease in that window exceeds a defined power threshold, then a ramp (either an up- or down-ramp) will be classified in that window. Up- and down-ramp events are allowed to happen in the same window of time. However, a large down-ramp interrupted by a small up-ramp (such as the up-ramp that occurs from hours 8-9 in the ramp window from hours 7-10 in Fig.2) will still be classified as a down-ramp as long as the overall down-ramp meets the power threshold criteria.

We added an additional sentence to clarify how a ramp is classified if a small interruption in the ramp direction happens.

L183: If a small up-ramp (down-ramp) interrupts an overall large down-ramp (up-ramp), the ramp will still be classified as a down-ramp (up-ramp) as long as the large ramp meets the power threshold criteria.

3.  The description of the fitting approach for the predictive distributions (Section 3.2) is also a bit unclear. As far as I understand, ordinary least squares regression is used to fit the transformed observations to the forecasts (and the forecasts to the seasonal cos and sin terms). Forecasts from this OLS model and its residual standard deviation are then used to derive parameters of truncated normal, truncated logistic or gamma distributions. How are the OLS outputs related to the distribution parameters? Did you simply use the method of moments? Some- how OLS assumes the residuals to follow a normal distribution and therefore this approach, if I understood it correctly, seems a bit weird. Why are you not using a distributional regression (i.e. EMOS like) approach where you

directly fit the distribution parameters as functions of the NWP forecasts and use a maximum likelihood or minimum CRPS framework?

Correct, we first power-transform the data to remove heteroscedasticity caused by an increase of uncertainty with increasing wind speeds, then used OLS to fit a model for the seasonal cycle and removed the seasonal cycle. After this data transformation the data are approximately homoscedastic, and while the residuals are not perfectly Gaussian, they are reasonably close to Gaussian to justify an OLS fit. In order to match the OLS parameters (conditional mean and standard deviation) to the distribution parameters of the gamma distribution we use a method of moments. For the truncated Gaussian and truncated logistic distribution we do not use an exact method of moments but instead matched the OLS parameters with the moments of the respective untruncated distributions. For 80 m wind speeds this approximation seemed justified because observations are sufficiently far away from zero for the truncation to be negligible (see Fig. 3b).

The advantage of this approach is its simplicity. It only requires standard OLS regression while still accounting for all sources of heteroscedasticity. An EMOS-like approach would be required and more adequate if some of the heteroscedasticity were explained by an additional predictor (e.g. ensemble spread). To address the increase of wind speed uncertainty with expected wind speed magnitude via EMOS, a more complex, parametric model for the standard deviation of the predictive distribution would be required, and we therefore chose the alternative modeling approach and addressed this issue via data transformation.

We added some additional detail to Section 3.2 to make our rationale clearer.

**Minor comments:**

1. L70, why is it a disadvantage to use nonparametric methods.

We reworded our statement so that nonparametric methods are not perceived as a disadvantage (L69). We also added a sentence that highlights the benefits of using a parametric approach for this study (L76).

2. Section 2.2: What are the lead times that are regarded. 1-12h? How long does it take the HRRR model to compute these forecasts and when do they become available? I.e., if the computation takes more than 1 hour the 1h forecast is not very useful.

For our analysis, we consider lead times 1-12 h from initialization. Some spin-up time is required, although this time is reduced by the digital diabatic filter initialization (DDFI), which acts to remove gravity waves excited by the prior data assimilation process. From initialization, the model can sometimes require about an hour to stabilize. We included the first hour of lead time as likely would be done in an operational setting and to include more valuable sliding windows in the analysis to look for ramp events. For the ramp calculations,

several other lead times are considered in the window of time in which we search for ramp events, so we do not expect the inclusion of the first lead time to affect the overall skill of the methods.

3. Caption of Figure 2: I think a window size of h=4 not h=3h is shown

Thank you for this observation. However, only three hours are considered in the window size. For instance, in Fig. 2a, hours 2-3, 3-4, and 4-5 are the hours used in the window (gray, shaded region). There are four data points used, but those equate to three hours.

4. P9L219: Not only the red dots but also black dots show the annual cycle.

That is correct. We modified the sentence to reflect that the black dots show the annual cycle too.

L246 now reads: The scatter in the red and black regression dots in Fig.3c illustrates how the annual cycle influences the regression; depending on the time of year, the transformation value can be different because of the annual cycle.

5. P12-13: In the description of the Gaussian copula approach, it should be clarified that the exponential covariance model (ECM) is an estimate of the covariances and that random number generation is based on a multivariate Gaussian distribution with these estimates as covariance parameter.

L336-339 now reads: An appropriate value for $v$ is selected empirically so that the resultant ECM for a given value of $v$ most resembles the decay of the empirical correlation values (Appendix A). A covariance matrix based on the ECM and the estimated value of $v$ is then set up and employed to randomly generate scenarios of multivariate Gaussian-distributed values.

6. Section 3.3.3: I expect η to be a quite important parameter and its optimal value clearly depends on the size of the data set. How did you select η = 50 and did you test the effect of this selection on the results?

The value of η is a compromise between representing the distribution well and making sure that there are a sufficient number of historical scenarios to select from. We tested the impact of selecting 20 and 50 scenarios on the results earlier on in the research process (not shown), but did not see a significant difference between the two selections for the Schaake Shuffle methods. We opted to move forward with η = 50 scenarios, which is a typical ensemble size used for operational purposes. Additionally, for the StSS method, it is advantageous to select scenarios that are still within the same season of the forecast day, so a much larger η may penalize the StSS method over the MDSS methods. The Gaussian Copula method needs more scenarios to represent the distribution well, because it uses random sampling rather than sampling the marginal distributions systematically by a set of quantiles. This characteristic is why we allowed 1000 scenarios for the GC method.

7. P19,L451-452: "On the other hand, we have shown that systematic over- and under-forecasting biases can be reduced with statistical post-processing". I did not find any results showing this in the manuscript.

We have changed the sentence at L513 to now read: However, we will show that systematic over- and under-forecasting biases in the climatological frequency of ramp events can be reduced with statistical post-processing (see Fig. 9).

8. Last paragraph on page 24: I do not fully agree that StSS is more competitive than MDSS and MDSS+. The differences mostly do not seem to be significant and e.g., for down-ramps at PNW StSS actually performs worse.

We rephrased this sentence to be more consistent with the results in figures 10 and 11.

L646: The most surprising result from the analyses is that the MDSS and MDSS+ methods are not overall significantly better than the StSS method despite their preferential selection of historical scenarios.

9. Caption of Figure 11: There are no Gaussian copula results shown

Thank you for catching this mistake. We corrected the caption.

10. Summary and discussion: It could be summarized here again, why multivariate methods are required to predict ramp events.

We added the following sentence to the summary and conclusion section:

L792: Because ramp events require simultaneous forecasts of multiple forecast lead times, multivariate statistical methods are a necessity for accurate ramp prediction.

**Responses to reviewer, E. Simon:**

The research work utilizes operational HRRRv2 NWP data, along with four methods of statistical post processing to transform deterministic wind speed forecasts into hourly probabilistic up and down-ramp predictions over sliding windows for two sites with complex terrain. The article is well written and is presented clearly and concisely. Ramp forecasting is a very important field which can lead to significant reductions in wind power balancing costs, and I believe that this original research work makes a valuable contribution to the topic. Therefore I recommend to the editor that this article is accepted for publication. The scientific approach is clear and well thought out, and the methods are presented in a reproducible and logical manner.

I have a few points that I ask the authors to consider for the final manuscript:

1. The observations themselves also have underlying uncertainties. Has the instrumentation been recently calibrated? You can likely disregard this (but still good to mention) since the time averaging will reduce the statistical uncertainty (type A)

   NREL is accredited by the American Association for Laboratory Accreditation, so the instruments meet requirements to ensure quality data. The M5 tower instruments were calibrated before installation and were recalibrated as required.

   The PNW tower is owned by a private wind power company, but the authors have verified that the instruments were calibrated upon installation and if needed afterwards.

2. How far are both sites from the HRRR model grid point? If the resolution is 3km, the closest point may not be representative (especially in complex terrain). Including a map with the mast locations and grid points would be a good addition.

   We added a figure (Fig. 1) showing the M5 tower location and the neighboring grid points. We were not permitted to show the PNW location, but this area is in less complex terrain than at the M5 site. The PNW tower is located in the Oregon side of the Columbia River Gorge, within 20 km of the river, between Wasco, OR and Boardman, OR.

At L136: We also added text describing the distance between the model grid point used and the actual tower locations.

3. A comparison of these methods with a baseline (e.g. persistence or climatology) would strengthen the results

We used the climatological probabilities of occurrence of ramp events as the reference/baseline forecasts in our Brier skill score calculations. We added more detail in Section 4.2 to describe why persistence may not be the best choice of a baseline forecast for ramp events.

L588-591: Climatological probabilities of occurrence of up- and down-ramp events with a particular $\xi$ and $h$ are used as the reference forecast. Persistence forecasts are another commonly-used baseline for wind power forecasting, but because ramp events can change magnitude and even direction in a short period of time, persistence is often not a practical estimate of ramp events.

4. Is there a reason to not first evaluate at a site with less complex flow, where the local observations are better correlated with model predictions? Especially since these methods have not been demonstrated before

This is a good point. We wanted to investigate wind power forecasts starting from the NOAA HRRR model, so we were geographically limited to the US. Additionally, we wanted to use the most updated version of the model. The corresponding observations required for statistical post processing needed to overlap the dates of the model availability and also had to be available for more than a year. Because of this criteria, we were only able to find tall tower data at these two sites. Now that we know the complexity of the site has so much of an impact on the ramp forecasts, it would be interesting to analyze these methods for less complex sites and for other numerical models. We mention these ideas as next directions for this research in the concluding paragraph.

5. It isn't entirely clear, since you say that M5 obs are averaged to 10 minutes, while PNW are averaged to 1 minute. Is that only raw data and then they are both averaged again later to 1-hour?

The raw tower data were available at 1 minute and 10 minute averages for the PNW and M5 towers, respectively. Some spiking issues were present for the raw 20Hz data from the M5 tower, but were not a problem in the available 10-min average data. Since the HRRR model output is instantaneous, we did not further average the tower data to one hour. We looked at 1-hr discretized model data, but we did not average any data over an hour. We made sure that the times of the observations matched the time of the model output (i.e, at the beginning of each hour). Some averaging of the observations is desired because although the model output is valid at the output time, it represents the grid cell, so some smoothing of the model data occurs.

We added more detail in the text to clarify this point. L139: The HRRR output is instantaneous at every hour, but because it represents a 3-km grid cell, comparing this output to averaged observations is preferred.

6.  Do you expect any improvements with the HRRRv3 model changes?

We can expect some improved forecasts of wind shear in the lowest 100 m and below hub height. Although many other improvements are expected, the wind speeds at hub height may not see a significant improvement. In regions of complex terrain, the addition of "geometric diffusion" in HRRRv3 may slightly improve wind variables as it performs horizontal mixing along Cartesian coordinates opposed to sigma coordinates. We do not expect significant improvements in the ramp results if the analyses were performed with HRRRv3.

7.  A note about processing time and ability to run these forecasts operationally would be useful information to provide

We added the following details to the conclusion section:

L859-865: The processing time for these methods is practical for real-time forecasting. Most of the processing does not even need to be run in real time. Only the multivariate methods need to be run in real time to generate probabilistic forecasts of ramp events. The Gaussian copula method is nearly instantaneous to compute, because it uses a random generator to produce scenarios. The MDSS and MDSS+ methods require the most time to process as they need to search through historical scenarios for the best matches to the current forecast. Nevertheless, they only required approximately three seconds to find 50 historical scenarios for one forecast day and initialization time. However, more time will be required to process the MDSS and MDSS+ methods for larger historical datasets.

8.  Skill scores for the M5 site using synthetic data are not shown (Figure 11 only for PNW). Are the results similar?

We technically did not generate synthetic data and calculate skill scores for the PNW site either. The caption for Figure 12 was confusing, so we corrected it to be more precise. We actually generated synthetic data based on higher correlation coefficients between modeled and observed wind ramps than were found using real data for the PNW site (i.e., correlations coefficients of up- and down-ramp events in Fig. 8). We used the correlation coefficients calculated from the PNW site as a bench mark and then generated synthetic data based on if the forecasts were much better (i.e., much higher correlation coefficients between synthetic forecasts and observations). The correlation coefficient between synthetic modeled and observed wind speed ramp events (calculated in the same way as Fig. 8 for the PNW and M5 correlation coefficients) was on average 0.72. This is much higher than the 0.23-0.37 values for the correlation coefficients calculated using the real data. The benefit of generating the synthetic data is that we can generate longer and better quality

forecasts. These better forecasts in turn allow us to really tell the differences between the Schaake Shuffle methods (i.e., the spread of the box plots in Fig. 12 are not as wide).

We also added this explanation of the synthetic data to the text:

L701: The synthetic forecasts were purposefully generated to be better forecasts than that of the real data so that differences among the different Schaake Shuffle methods would be more apparent.

9. The caption on pg. 26 should be on the same page as the figure. Same with pg. 27 to reduce whitespace

Thank you. We corrected this.